# Lightspeed Geometric Dataset Distance via Sliced Optimal Transport

Khai Nguyen [*1]   Hai Nguyen [*2]   Tuan Pham [1]   Nhat Ho [1]

## Abstract

We introduce *sliced optimal transport dataset distance* (s-OTDD), a model-agnostic, embedding-agnostic approach for dataset comparison that requires no training, is robust to variations in the number of classes, and can handle disjoint label sets. The core innovation is Moment Transform Projection (MTP), which maps a label, represented as a distribution over features, to a real number. Using MTP, we derive a data point projection that transforms datasets into one-dimensional distributions. The s-OTDD is defined as the expected Wasserstein distance between the projected distributions, with respect to random projection parameters. Leveraging the closed form solution of one-dimensional optimal transport, s-OTDD achieves (near-)linear computational complexity in the number of data points and feature dimensions and is independent of the number of classes. With its geometrically meaningful projection, s-OTDD strongly correlates with the optimal transport dataset distance while being more efficient than existing dataset discrepancy measures. Moreover, it correlates well with the performance gap in transfer learning and classification accuracy in data augmentation.

## 1. Introduction

Dataset distances provide a powerful framework for comparing datasets based on their underlying structures, distributions, or content. These measures are essential in applications where understanding the relationships between datasets drives decision-making, such as assessing data quality, detecting distributional shifts, or quantifying biases. They play a critical role in machine learning workflows, enabling tasks like domain adaptation, transfer learning,

continual learning, and fairness evaluation. Additionally, dataset distances are valuable in emerging areas such as synthetic data evaluation, 3D shape comparison, and federated learning, where comparing heterogeneous data distributions is fundamental. By capturing meaningful similarities and differences between datasets, these measures facilitate data-driven insights, enhance model robustness, and support novel applications across diverse fields.

A common approach to comparing datasets relies on proxies, such as analyzing the learning curves of a predefined model (Leite & Brazdil, 2005; Gao & Chaudhari, 2021) or examining its optimal parameters (Achille et al., 2019; Khodak et al., 2019) on a given task. Another strategy involves making strong assumptions about the similarity or co-occurrence of labels between datasets (Tran et al., 2019). However, these methods often lack theoretical guarantees, are heavily dependent on the choice of the probe model, and require training the model to completion (e.g., to identify optimal parameters) for each dataset under comparison. To address limitations of previous approaches, model-agnostic approaches are developed. These methods often assess task similarity based on the similarity between joint or conditional input-output distributions, occasionally incorporating the loss function as well. Principled notions of domain discrepancy (Ben-David et al., 2006; Mansour et al., 2009) provide a more rigorous foundation but are frequently impractical due to computational intractability or poor scalability to large datasets.

More recently, approaches based on Optimal Transport (Villani, 2008; Peyré & Cuturi, 2020) (OT) have demonstrated promise in modeling dataset similarities (Alvarez-Melis & Fusi, 2020; Tan et al., 2021). Authors in (Alvarez-Melis & Fusi, 2020) introduced optimal transport dataset distance (OTDD) which is based on a hierarchical OT framework drawing from (Yurochkin et al., 2019). In this approach, an inner OT problem calculates the label distance between the class-conditional distributions of two supervised learning tasks. This label distance is then integrated into the transportation cost of an outer OT problem, yielding a dataset distance that accounts for both sample and label discrepancies, and can handle disjoint sets of labels from different datasets. Meanwhile, authors in (Tan et al., 2021) treat the optimal transport plan between the input distributions of datasets as a joint probability distribution and leverage con-

*Equal contribution [1]Department of Statistics and Data Sciences, University of Texas at Austin, Texas, USA [2]Qualcomm AI Research, Qualcomm Vietnam Company Limited. Correspondence to: Khai Nguyen <khainb@utexas.edu>.

*Proceedings of the 42$^{nd}$ International Conference on Machine Learning*, Vancouver, Canada. PMLR 267, 2025. Copyright 2025 by the author(s).

ditional entropy to quantify the difference between datasets.

A key limitation of OT-based approaches is their high computational complexity. These methods require pairwise OT (or entropy-regularized OT) computations across datasets and classes. This process can be prohibitively expensive, especially for applications that demand frequent dataset similarity evaluations. Solving optimal transport incurs a time complexity of $\mathcal{O}(n^3 \log n)$ (Peyré & Cuturi, 2020) and a space complexity of $\mathcal{O}(n^2)$, where $n$ is the maximum number of realizations in the two input distributions. Using entropic regularization (Cuturi, 2013) for approximate solutions reduces the complexity to $\mathcal{O}(n^2 \log n/\epsilon^3)$ (Altschuler et al., 2017), with $\epsilon$ denoting the accuracy level. To enhance efficiency, (Liu et al., 2025) proposes using multidimensional scaling (MDS) (Cox & Cox, 2000) to embed class labels as vectors while preserving the Wasserstein distance geometry for label distances. The authors then leverage a reference distribution to form Wasserstein embeddings of datasets (Wang et al., 2013; Kolouri et al., 2021), termed Wasserstein Task Embedding (WTE). Although WTE is more efficient than OTDD, performing MDS and Wasserstein embedding remains costly due to the need for solving optimal transport. Additionally, WTE is not a valid metric.

Sliced optimal transport or sliced Wasserstein (SW) distance (Bonneel et al., 2015; Rabin et al., 2012) is a well-known approach to obtain scalable optimal transport distance between distributions. The SW relies on random projections to exploit the closed-form of Wasserstein distance in one-dimension with $\mathcal{O}(n \log n)$ in time complexity and $\mathcal{O}(n)$ in space complexity. The challenge in using SW as a dataset distance is that carrying out projection for class labels is non-trivial. Authors in (Bonet et al., 2024) bypass the problem by using MDS to embed class labels into vectors on a chosen manifold, then utilize SW on product of manifolds i.e., Cartan-Hadamard Sliced-Wasserstein (CHSW), as the dataset distance. Nevertheless, conducing MDS as WTE leads to extra computation which has quadratic computation in terms of the number of classes that is undesirable when having a large number of classes.

Our goal is to develop a model-agnostic sliced optimal transport distance for comparing datasets, which is also embedding-agnostic, requiring no additional preprocessing. Furthermore, we aim for the distance to achieve (near-)linear computational complexity with respect to the number of data points and feature dimensions, while being robust to the number of classes in the datasets. To this end, we propose a novel approach for projecting a data point-comprising a feature and a label-onto a real number. This projection includes an innovative label (class) projection, which maps a label, represented as a conditional distribution over features (as in OTDD), to a scalar.

**Contribution**. In summary, our contributions are three-fold:

1. We propose Moment Transform Projection (MTP), which maps a label, a distribution over high-dimensional features, to a single scalar. MTP first projects the distribution of interest onto one dimension using a feature projection, then computes a scaled moment to convert the projected one-dimensional distribution into a scalar. We prove that MTP is injective under certain regularity assumptions of the underlying dataset distributions. Using MTP, we derive data point projection that combines the label projections from multiple MTPs with a feature projection to obtain a one-dimensional representation of a data point. By connecting data point projection to the hierarchical hybrid projection approach (Nguyen & Ho, 2024), we show that data point projection is injective, given the injectivity of the MTP.

2. We propose the sliced optimal transport dataset distance (s-OTDD), the first formal dataset distance based on sliced optimal transport. The s-OTDD is defined as the expected Wasserstein distance between the projected one-dimensional distributions of two input datasets, where the projections are determined by random parameters. We prove that the s-OTDD is a valid distance on the space of distributions over the product of the feature space and the space of all distributions on the feature space. Moreover, we dicuss in detail the computational properties of the s-OTDD including approximation error, computational complexities, and other computational benefits.

3. We show that the s-OTDD exhibits a strong correlation with the OTDD when comparing subsets from MNIST and CIFAR10, while being faster than existing competitors. Moreover, we observe that it correlates well with the performance gap in transfer learning across various datasets and modalities, including NIST datasets for images, as well as AG News, DBPedia, Yelp Reviews, Amazon Reviews, and Yahoo Answers for text. Finally, we also find that the s-OTDD correlates well with classification accuracy in data augmentation on CIFAR10 and Tiny-ImageNet.

**Organization.** We begin by reviewing preliminaries on optimal transport dataset distance in Section 2. Next, we present the proposed s-OTDD and its key contributions in Section 3. Experiments demonstrating the favorable performance of the s-OTDD are provided in Section 4. Finally, we conclude in Section 5 and defer proofs of key results and additional materials to the Appendices.

**Notations.** We denote $\mathcal{P}(\mathcal{X})$ as the set of all distributions on the set $\mathcal{X}$. We denote $\mathbb{R}^d$ as the set of real numbers in $d > 0$ dimensions, $\mathbb{N} = 1, 2, \ldots, \infty$ as the set of natural numbers, and $\mathbb{S}^{d-1}$ as the unit hypersphere in $d > 1$ dimensions. For a dataset $\mathcal{D} = (x_1, y_1), \ldots, (x_n, y_n)$, we define its empirical distribution as $P_{\mathcal{D}} = \frac{1}{n} \sum_{i=1}^{n} \delta_{(x_i, y_i)}$. We denote $[[n]]$ as the set $\{1, \ldots, n\}$. For any two sequences $a_n$ and $b_n$, the notation $a_n = \mathcal{O}(b_n)$ means that $a_n \leq C b_n$ for all $n \geq 1$ for a constant $C$. Lastly, $\lambda!$ denotes the factorial of $\lambda > 0$.

## 2. Preliminaries

In this section, we review the definition of Wasserstein distance, sliced Wasserstein distance, and their computational properties, and restate the optimal transport dataset distance.

**Wasserstein Distance.** The Wasserstein-$p$ distance (Villani, 2008; Peyré & Cuturi, 2020) ($p \geq 1$) between two distributions $\mu \in \mathcal{P}(\mathcal{X})$ and $\nu \in \mathcal{P}(\mathcal{X})$, where $\mathcal{X}$ are subsets of $\mathbb{R}^d$ and has a ground metric $d_{\mathcal{X}} : \mathcal{X} \times \mathcal{X} \to \mathbb{R}^+$, is defined as:

$$\mathrm{W}_p^p(\mu, \nu) := \inf_{\pi \in \Pi(\mu, \nu)} \int_{\mathcal{X} \times \mathcal{X}} d_{\mathcal{X}}(x, x')^p d\pi(x, x'), \quad (1)$$

where $\Pi(\mu, \nu)$ is the set of all possible transportation plan i.e., all joint distributions $\pi(x, x')$ such that $\pi(x, \mathcal{X}) = \mu(x)$ and $\pi(\mathcal{X}, x') = \nu(x')$. When $\mu$ and $\nu$ are discrete i.e., $\mu = \sum_{i=1}^n \alpha_i \delta_{x_i}$ and $\nu = \sum_{j=1}^m \beta_j \delta_{x'_j}$ with $\sum_{i=1}^n \alpha_i = \sum_{j=1}^m \beta_j = 1$ ($\alpha_i > 0 \forall i$, and $\beta_j > 0 \forall j$), we have:

$$\mathrm{W}_p^p(\mu, \nu) := \min_{\gamma \in \Gamma(\boldsymbol{\alpha}, \boldsymbol{\beta})} \sum_{i=1}^n \sum_{j=1}^m \gamma_{ij} d_{\mathcal{X}}^p(x_i, x'_j), \quad (2)$$

where $\Gamma(\boldsymbol{\alpha}, \boldsymbol{\beta}) = \{\gamma \in \mathbb{R}_+^{n \times m} \mid \gamma \mathbf{1} = \boldsymbol{\beta}, \gamma^\top \mathbf{1} = \boldsymbol{\alpha}\}$, $\boldsymbol{\alpha} = (\alpha_1, \ldots, \alpha_n)$, and $\boldsymbol{\beta} = (\beta_1, \ldots, \beta_n)$. The time complexity and the space complexity of the Wasserstein distance is $\mathcal{O}(n^3 \log n)$ (Peyré & Cuturi, 2020) and $\mathcal{O}(n^2)$ in turn which are very expensive. Using entropic regularization reduces the complexity to $\mathcal{O}(n^2 \log n / \epsilon^3)$ with $\epsilon$ denoting the accuracy level. To avoid such quadratic complexity, sliced Wasserstein is proposed as an alternative solution.

**Sliced Wasserstein distance.** The sliced Wasserstein (SW) distance is motivated from the closed-form solution of the one-dimensional Wasserstein distance with ground metric $d_{\mathcal{X}}(x, x') = h(x - x')$ with $\mathcal{X} \subset \mathbb{R}$ and $h$ is a strictly convex function:

$$\mathrm{W}_p^p(\mu, \nu) = \int_0^1 d_{\mathcal{X}}\left(F_\mu^{-1}(z), F_\nu^{-1}(z)\right)^p dz, \quad (3)$$

where $F_\mu^{-1}$ and $F_\nu^{-1}$ are inverse CDF of $\mu$ and $\nu$ respectively. When $\mu$ and $\nu$ are discrete with at most $n$ supports, the time complexity and the space complexity for computing the closed-form are $\mathcal{O}(n \log n)$ (Peyré & Cuturi, 2020) and $\mathcal{O}(n)$ respectively. To exploit the closed-form, the SW distance relies on random projections. For $p \geq 1$, the SW distance (Bonneel et al., 2015) of $p$-th order between two distributions $\mu \in \mathcal{P}(\mathcal{X})$ and $\nu \in \mathcal{P}(\mathcal{X})$ with $\mathcal{X} \subset \mathbb{R}^d$ is defined as follow:

$$\mathrm{SW}_p^p(\mu, \nu) = \mathbb{E}_{\theta \sim \mathcal{U}(\mathbb{S}^{d-1})}[\mathrm{W}_p^p(\mathcal{R}_\theta \sharp \mu, \mathcal{R}_\theta \sharp \nu)], \quad (4)$$

where $\mathcal{R}_\theta \sharp \mu$ and $\mathcal{R}_\theta \sharp \nu$ are the one-dimensional push-forward distributions of $\mu$ and $\nu$ through the function $\mathcal{R}_\theta(x) = \theta^\top x$ (derived from Radon Transform (RT) (Helgason, 2011)), and $\mathcal{U}(\mathbb{S}^{d-1})$ is the uniform distribution over

the unit hypersphere in $d$ dimensions. In addition to the computational benefit, SW has been widely known for its low sample complexity (Manole et al., 2022; Nadjahi et al., 2020; Nietert et al., 2022; Boedihardjo, 2025).

**Optimal Transport dataset distance.** We are given two datasets $\mathcal{D}_1 = \{(x_i, y_i)\}_{i=1}^n$ and $\mathcal{D}_2 = \{(x'_j, y'_j)\}_{j=1}^m$, where $(x_i, y_i), (x'_j, y'_j) \in \mathcal{X} \times \mathcal{Y}, \forall i \in [[n]], \forall j \in [[m]]$ with $\mathcal{X}$ is the space of features and $\mathcal{Y}$ is the space of labels. We assume that we know the support distance on the space of features $\mathcal{X}$ i.e., $d_{\mathcal{X}} : \mathcal{X} \times \mathcal{X} \to \mathbb{R}^+$ e.g., Euclidean distance. In contrast to the mild assumption of having $d_{\mathcal{X}}$, we rarely know the distance on the space of labels $\mathcal{Y}$ e.g., we cannot tell if the label "dog" is closer to the label "cat" than to the label "bird". As a solution, authors in (Alvarez-Melis & Fusi, 2020) propose to map a label into a distribution over features, then use distances between distributions as the proxy for the distance between labels. In particular, for the first dataset, we assume that $(x_1, y_1), \ldots, (x_n, y_n) \sim q(X, Y)$ where $q(X, Y)$ is an unknown joint distribution. After that, we can define $q_y(X) = q(X|Y = y)$ as the conditional distribution over features given the label $y$. Similarly, for the second dataset, we can obtain $q_{y'}(X')$ for a label $y'$. Therefore, the distance between labels can be defined as:

$$d_{\mathcal{Y}}(y, y') = \mathbb{D}(q_y(X), q_{y'}(X')), \quad (5)$$

where $\mathbb{D}$ is a distance between two distributions. In practice, we observe $q_y(X)$ and $q_{y'}(X')$ in empirical forms, hence, optimal transport distances naturally serve as a measure for $\mathbb{D}$. Authors in (Alvarez-Melis & Fusi, 2020) suggest using the Wasserstein distance i.e., $d_{\mathcal{Y}}(y, y') = \mathrm{W}_p(q_y(X), q_{y'}(X'))$, where we abuse the notation of the Wasserstein distance between two pdfs as the Wasserstein distance between the two distributions. Due to the expensive computational complexities of the Wasserstein distance, authors in (Alvarez-Melis & Fusi, 2020) propose to approximate two distributions by two multivariate Gaussians, then use the closed-form solution of Wasserstein-2 distance as the label distance. Nevertheless, the approximation can only capture the first two moments of the two distributions, hence, the resulting comparison might not be accurate. After having the label distance, the optimal transport dataset distance can be defined as:

$$OTDD(\mathcal{D}_1, \mathcal{D}_2)$$
$$= \min_{\gamma \in \Gamma(\frac{1}{n}, \frac{1}{m})} \sum_{i=1}^n \sum_{j=1}^m \gamma_{ij} d((x_i, y_i), (x'_j, y'_j)), \quad (6)$$

which is the Wasserstein distance with the ground metric: $d^p((x_i, y_i), (x'_j, y'_j)) = d_{\mathcal{X}}^p(x_i, x'_j) + d_{\mathcal{Y}}^p(y_i, y'_j)$, which is the combination of the feature distance and the label distance. As long as the feature distance and the label distance are valid metrics, the OTTD is a valid metric on the product space of $\mathcal{X} \times \mathcal{P}(\mathcal{X})$ (Alvarez-Melis & Fusi, 2020).

# 3. Sliced Optimal Transport Dataset Distances

While the OTDD is a natural distance between two datasets, it inherits the computational challenges of the Wasserstein distance. Let $n$ denote the maximum number of data points in the two datasets, $c$ the maximum number of classes, and $n_{\max}$ the maximum number of samples per class. The time complexity of OTDD is $\mathcal{O}(n^3 \log n + c^2(n_{max}^3 \log n_{\max} + d))$, and the memory complexity is $\mathcal{O}(n^2 + c^2)$. When using a Gaussian approximation for the label distribution, the time complexity becomes $\mathcal{O}(n^3 \log n + c^2(d^3 + n_{\max}d^2))$. As a result, OTDD may not be scalable for large datasets. To address this, we aim to develop a sliced optimal transport version of OTDD that leverages the one-dimensional closed-form of the Wasserstein distance. This requires developing a novel approach to project a *data point* onto a single *scalar*.

## 3.1. Label Projection

As mentioned in Section 2, a label is represented as a distribution over the feature space $\mathcal{X}$. Given a label $y$, we would like to map the label distribution $q_y$ to a scalar. To achieve our goal, there are two steps: projecting $q_y$ to one-dimension through feature projection, and transforming the one-dimensional projection of $q_y$ into a scalar.

**Feature Projection.** Since a label is treated as a distribution over the feature space, we can project a label to an one-dimensional distribution through a feature projection i.e., a mapping from $\mathcal{FP}_\theta : \mathcal{X} \to \mathbb{R}$ with the projection parameter $\theta$ belongs to a projection space $\Theta$. We can choose any feature projection methods based on the prior knowledge of the feature space $\mathcal{X}$ e.g., Euclidean space (Bonneel et al., 2015), images (Nguyen & Ho, 2022), functions (Garrett et al., 2024), spherical space (Tran et al., 2024; Quellmalz et al., 2023; 2024), hyperbolic space (Bonet et al., 2023), manifolds (Bonet et al., 2024; Nguyen & Mueller, 2024), and so on. It is worth noting that a feature projection is injective if $\mathcal{FP}_\theta \sharp \mu_1 = \mathcal{FP}_\theta \sharp \mu_2$ for all $\theta \in \Theta$ then $\mu_1 = \mu_2$ for any $\mu_1, \mu_2 \in \mathcal{P}(\mathcal{X})$. The injectivity of the projection is vital to preserve the distributional metricity.

**Scaled Moment.** We now discuss how to map a one-dimensional distribution into a scalar. More importantly, we want the transformation to be injective. To design such transformation, we rely on scaled moments.

**Definition 1.** Given a distribution $\mu \in \mathcal{P}(\mathbb{R})$ with the density function $f_\mu$ and a set $\Lambda \subset \mathbb{N}$ such that $\int_\mathbb{R} \frac{x^\lambda}{\lambda!} f_\mu(x)dx < \infty$ for all $\lambda \in \Lambda$, the $\lambda$-th scaled moment of $\mu$ is defined as:

$$\mathcal{SM}_\lambda(\mu) = \int_\mathbb{R} \frac{x^\lambda}{\lambda!} f_\mu(x)dx \qquad (7)$$

The difference between the scaled moment and the conventional moment is that the scaled moment is scaled by the factorial function of the order $\lambda$. The scaling is for avoiding

exploding in value of high moments.

**Moment Transform Projection.** After the feature projection, we obtain an one-dimensional projection of the label distribution $\mathcal{FP}_\theta \sharp q_y$. We then can obtain the scaled moment of $\mathcal{FP}_\theta \sharp q_y$ as the final desired scalar. Before giving the formal definition, we first state the following assumption.

**Assumption 1** (Existence of projected scaled moments)**.** A distribution $\mu \in \mathcal{P}(\mathbb{R}^d)$ with the density function $f_\mu$ has all projected $\lambda$-th scaled moments ($\lambda \in \Lambda \subset \mathbb{N}$) given a feature projection $\mathcal{FP}_\theta$ if $\int_{\mathbb{R}^d} \frac{(\mathcal{FP}_\theta(x))^\lambda}{\lambda!} f_\mu(x)dx < \infty$ for all $\theta \in \mathbb{S}^{d-1}$ and $\lambda \in \Lambda$.

**Definition 2.** Given a feature projection $\mathcal{FP}_\theta : \mathcal{X} \to \mathbb{R}$ and a set $\Lambda \subset \mathbb{N}$, a distribution $\mu \in \mathcal{P}(\mathbb{R}^d)$ ($d > 1$) that satisfies Assumption 1, the Moment Transform projection $\mathcal{MTP}_{\lambda,\theta} : \mathcal{P}(\mathbb{R}^d) \to \mathbb{R}$, with $\theta \in \mathbb{S}^{d-1}$ and $\lambda \in \Lambda$, is defined as follows:

$$\mathcal{MTP}_{\lambda,\theta}(\mu) = \mathcal{SM}_\lambda(\mathcal{FP}_\theta \sharp \mu)$$
$$= \int_{\mathbb{R}^d} \frac{(\mathcal{FP}_\theta(x))^\lambda}{\lambda!} f_\mu(x)dx. \qquad (8)$$

When $\mu$ is an empirical distribution i.e., $\mu = \frac{1}{n}\sum_{i=1}^n \delta_{x_i}$, the MTP projection of $\mu$ given $\lambda$ and a feature projection $\mathcal{FP}_\theta$ is $\mathcal{MTP}_{\lambda,\theta} = \frac{1}{n}\sum_{i=1}^n \frac{(\mathcal{FP}_\theta(x))^\lambda}{\lambda!}$ . We now discuss the distributionally injectivity of the MTP.

**Proposition 1.** *For $\mu, \nu \in \mathcal{P}(\mathbb{R}^d)$ and a injective feature projection $\mathcal{FP}_\theta$ and a set $\Lambda \subset \mathbb{N}$, $\mathcal{MTP}_{\lambda,\theta}(\mu) = \mathcal{MTP}_{\lambda,\theta}(\nu)$ for all $\theta \in \mathbb{S}^{d-1}$ and $\lambda \in \Lambda$ implies $\mu = \nu$ if either following conditions hold:*

*(1) $\mu$ and $\nu$ satisfies Assumption 1 with an infinite set $\Lambda = \mathbb{N}$ and moment generating functions (MGFs) of $\mathcal{FP}_\theta \sharp \mu$ and $\mathcal{FP}_\theta \sharp \nu$ exist for all $\theta \in \Theta$.*

*(2) $\mu$ and $\nu$ satisfies Assumption 1 with an finite set $\Lambda \subset \mathbb{N}$, and for all $\theta \in \Theta$, the projected Hankel matrices*

$$A_{\theta,\mu} = \begin{bmatrix} m_{\theta,\mu,0} & m_{\theta,\mu,1} & \cdots & m_{\theta,\mu,\lambda_{\max}} \\ m_{\theta,\mu,1} & m_{\theta,\mu,2} & \cdots & m_{\theta,\mu,0} \\ \vdots & \vdots & \ddots & \vdots \\ m_{\theta,\mu,\lambda_{\max}} & m_{\theta,\mu,0} & \cdots & m_{\theta,\mu,\lambda_{\max}-1} \end{bmatrix}$$

*with $m_{\theta,\mu,\lambda}$ is $\lambda$-th moment of $\mathcal{FP}_\theta \sharp \mu$ (similar with $m_{\theta,\nu,\lambda}$) are positive definite and $|m_{\theta,\mu,\lambda}| < CD^\lambda \lambda!$ and $|m_{\theta,\nu,\lambda}| < CD^\lambda \lambda!$ for some constants $C$ and $D$ for all $\lambda \in \Lambda$ .*

The proof of Proposition 1 is given in Appendix A.1 and is based on the Hamburger moment problem (Reed & Simon, 1975; Chihara, 2011).

## 3.2. Data Point Projection

With the proposed label projection, we can propose data point projection. We recall that a data point is a pair of a

feature and a label or equivalently a pair of a feature and a distribution over features, which is denoted as $(x, q_y) \in \mathcal{X} \times \mathcal{P}(\mathcal{X})$. For the feature domain, as discussed, we can use any feature projections based on the prior knowledge of the feature space, denoted as $\mathcal{FP}_\theta : \mathcal{X} \to \mathbb{R}$. For the label, we can use MTP defined in the Section 3.1. Now, we need to combine the outputs of the feature projection and the label projection to obtain the final scalar.

**Definition 3.** Given a data point $(x, q_y) \in \mathcal{X} \times \mathcal{P}(\mathcal{X})$ and $k \geq 1$, the data point projection can be defined as follows:

$$
\mathcal{DP}^k_{\psi,\theta,\lambda,\phi}(x, q_y) = \psi^{(1)} \mathcal{FP}_\theta(x)
$$
$$
+ \sum_{i=1}^{k} \psi^{(i+1)} \mathcal{MTP}_{\lambda^{(i)},\phi}(q_y), \quad (9)
$$

where $\psi = (\psi^{(1)}, \psi^{(2)}, \ldots, \psi^{(k+1)}) \in \mathbb{S}^k, \theta \in \Theta, \lambda = (\lambda^{(1)}, \ldots, \lambda^{(k)}) \in \Lambda^k, \phi \in \Phi$ with $\Theta$ and $\Phi$ are projection space of the feature projections.

Given a dataset $\mathcal{D} = \{(x_1, q_{y_1}), \ldots, (x_1, q_{y_n})\}$, we have the projected distribution of the corresponding empirical distribution through the data point projection is $\mathcal{DP}^k_{\psi,\theta,\lambda,\phi} \sharp P_\mathcal{D} = \frac{1}{n} \sum_{i=1}^{n} \delta_{\mathcal{DP}^k_{\psi,\theta,\lambda,\phi}(x_i, q_{y_i})}$. The proposed data point projection combines the outputs of $k \geq 1$ MTPs to obtain the final projection value. It follows the principle of hierarchical hybrid projection in (Nguyen & Ho, 2024) to retain the overall injectivity.

**Corollary 1.** *The data point projection is injective when the feature projection and the MTP is injective i.e., for $\mu, \nu \in \mathcal{X} \times \mathcal{P}(\mathcal{X})$ if $\mathcal{DP}^k_{\psi,\theta,\lambda,\phi} \sharp \mu = \mathcal{DP}^k_{\psi,\theta,\lambda,\phi} \sharp \nu$ for all $\psi \in \mathbb{S}, \theta \in \Theta, \lambda \in \Lambda \subset \mathbb{N}, \phi \in \Phi$.*

The Corollary 1 follows the fact that the data point projection is a composition of injective projections (Nguyen & Ho, 2024) i.e., the Radon Transform projection and the MTP.

It is worth noting that we can use the same value for $\theta$ and $\phi$ when we use a single feature projection to project feature and to construct label projections. This approach not only reduces memory consumption for storing projection parameters but also saves computation since we need to project features only once.

### 3.3. Sliced Optimal Transport Dataset Distance

With projections of data points, we can now introduce the sliced optimal transport dataset distance (s-OTDD).

**Definition 4.** Let $\mathcal{D}_1$ and $\mathcal{D}_2$ be the two given datasets, $P_{\mathcal{D}_1}$ and $P_{\mathcal{D}_2}$ be corresponding empirical distributions of $\mathcal{D}_1$ and $\mathcal{D}_2$ respectively, the sliced optimal transport dataset distance (s-OTDD) of order $p > 0$ is defined as follows:

$$
\text{s-OTDD}^p_p(\mathcal{D}_1, \mathcal{D}_2)
$$
$$
= \mathbb{E} \left[ \text{W}^p_p(\mathcal{DP}^k_{\psi,\theta,\lambda,\phi} \sharp P_{\mathcal{D}_1}, \mathcal{DP}^k_{\psi,\theta,\lambda,\phi} \sharp P_{\mathcal{D}_2}) \right], \quad (10)
$$

where the expectation is with respect to the random projection parameters $(\psi, \theta, \lambda, \phi) \sim \mathcal{U}(\mathbb{S}) \otimes \mathcal{U}(\mathbb{S}^{d-1}) \otimes \sigma(\Lambda^k) \otimes \mathcal{U}(\Phi)$ with $\phi \in \Phi$, and $\sigma(\Lambda^k)$ is the uniform distribution on $\Lambda^k$ when $\Lambda$ is finite or the product of zero-truncated Poisson distributions (Cohen, 1960) when $\Lambda$ is infinite.

**Proposition 2.** *The sliced optimal transport dataset distance s-OTDD$_p(\mathcal{D}_1, \mathcal{D}_2)$ defines a valid metric on $\mathcal{P}(\mathcal{X} \times \mathcal{P}(\mathcal{X}))$ the space of distributions over feature and label-distribution pairs if the data point projection is injective.*

Proof of Proposition 2 is given in Appendix A.2. The proposition strengthens the geometrical benefits of the s-OTDD.

**Numerical Approximation.** The expectation in s-OTDD is intractable, hence, numerical approximation such as Monte Carlo integration must be used. In particular, we sample $(\psi_1, \theta_1, \lambda_1, \phi_1), \ldots, (\psi_L, \theta_L, \lambda_L, \phi_L) \sim \mathcal{U}(\mathbb{S}) \otimes \mathcal{U}(\mathbb{S}^{d-1}) \otimes \sigma(\Lambda^k) \otimes \mathcal{U}(\Phi)$ with the number of projections $L > 0$, then we form the following estimation:

$$
\widehat{\text{s-OTDD}}^p_p(\mathcal{D}_1, \mathcal{D}_2; L) =
$$
$$
\frac{1}{L} \sum_{l=1}^{L} \text{W}^p_p(\mathcal{DP}^k_{\psi_l,\theta_l,\lambda_l,\phi_l} \sharp P_{\mathcal{D}_1}, \mathcal{DP}^k_{\psi_l,\theta_l,\lambda_l,\phi_l} \sharp P_{\mathcal{D}_2}). \quad (11)
$$

In practice, we do not know the number of moments. Hence, we can assume it to be infinite and use the product of zero-truncated Poisson distributions (Cohen, 1960) with some rate hyperparameters as $\sigma(\Lambda^k)$. Another approach is to cap the number of moments to be $\lambda_{max} \geq 1$ i.e., $\Lambda = \{1, 2, \ldots, \lambda_{max}\}$, then use the uniform distribution $\mathcal{U}(\Lambda^k)$ as $\sigma(\Lambda^k)$. We now discuss the approximation error of the s-OTDD when using the Monte Caro estimation.

**Proposition 3.** *For any $p \geq 1$, and two datasets $\mathcal{D}_1$ and $\mathcal{D}_2$, we have the following approximation error:*

$$
\mathbb{E} \left[ \left| \widehat{s\text{-}OTDD}^p_p(\mathcal{D}_1, \mathcal{D}_2; L) - s\text{-}OTDD^p_p(\mathcal{D}_1, \mathcal{D}_2) \right| \right] \leq
$$
$$
\frac{1}{\sqrt{L}} Var \left[ W^p_p(\mathcal{DP}^k_{\psi,\theta,\lambda,\phi} \sharp P_{\mathcal{D}_1}, \mathcal{DP}^k_{\psi,\theta,\lambda,\phi} \sharp P_{\mathcal{D}_2}) \right], \quad (12)
$$

*where the variance is with respect to $(\psi, \theta, \lambda, \phi) \sim \mathcal{U}(\mathbb{S}) \otimes \mathcal{U}(\mathbb{S}^{d-1}) \otimes \sigma(\Lambda^k) \otimes \mathcal{U}(\Phi)$.*

The proof of Proposition 3 is given in Appendix A.3. We can see that the approximation error reduce fast when increasing the number of projections $L$ i.e., $\mathcal{O}(L^{-1/2})$. It is worth noting that variance reduction can also be used (Nguyen & Ho, 2023; Nguyen et al., 2024; Leluc et al., 2024). We refer the read to Algorithm 1 in Appendix B for a detailed computational algorithm.

**Computational Complexities.** Using a feature projection with linear complexity in the number of dimensions $d$ (e.g., Radon transform, convolution, etc), the time complexity of s-OTDD is $\mathcal{O}(L(n \log n + dn))$ and the space complexity is

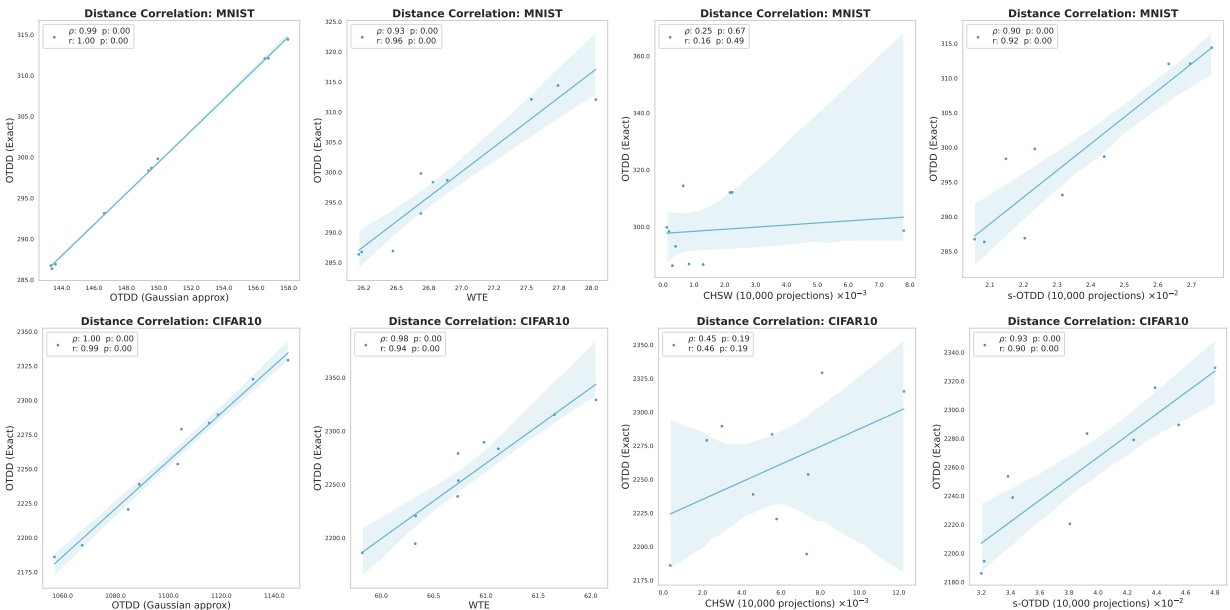

Figure 1: The figure shows distance correlation with OTDD (Exact) of OTDD (Gaussian approximation), WTE, CHSW, and s-OTDD.

$\mathcal{O}(L(d+n))$, where $n$ is the number of data points and $L$ is the number of projections. Interestingly, both time complexity and memory complexity do not depend on the number of classes $c$ and the maximum class size $n_{max}$ (good for imbalanced data). In a greater detail, the time complexity for applying projection for the class $i$ is $\mathcal{O}(Ln_i d)$ where $n_i$ is the size of the class $i$. As a result, the total time complexity for projection is $\mathcal{O}\left(L\sum_{i=1}^{k}n_i d\right) = \mathcal{O}(Lnd)$. After having the projections, solving one-dimensional Wasserstein distances costs $\mathcal{O}(Ln\log n)$ in time. For space complexity, $\mathcal{O}(L(d+n))$ is for storing projection parameters and projections. It is worth noting that the majority of computation of s-OTDD is for projecting and computing inverse CDFs of projected distributions of datasets, and such computation can be done independently for each dataset. After that, s-OTDD value can be obtained with linear complexity with the precomputed projected inverse CDFs. Therefore, s-OTDD is suitable for distributed and federated settings where datasets can be stored across machines.

## 4. Experiments

In Section 4.1, we conduct analysis for the s-OTDD in comparing subsets of the MNIST dataset (LeCun et al., 1998) and the CIFAR10 dataset (Krizhevsky et al., 2009). In particular, we analyze the correlations, which Spearman's rank correlation denoted by $\rho$ and Pearson correlation denoted by $r$, of the s-OTDD with OTDD. While prior work (Alvarez-Melis & Fusi, 2020) reports only Spearman's rank correlation, we also compute Pearson correlation for a more comprehensive analysis, though Spearman's correlation remains our primary metric to maintain comparability with existing

baselines. Furthermore, we compare the computational time of the proposed s-OTDD with existing dataset distance metrics and investigate its dependency on the number of projections and on the number of moments. In Section 4.2, we evaluate the correlations of the s-OTDD with the performance gap in transfer learning on image NIST datasets (Deng, 2012) and a diverse set of text datasets (Zhang et al., 2015) including AG News, DBPedia, Yelp Reviews (with both 5-way classification and binary polarity labels), Amazon Reviews (with both 5-way classification and binary polarity labels), and Yahoo Answers. Also, to demonstrate the robustness and scability of proposed method, we empirically validate the correlations between s-OTDD distance and transferability on large-scale dataset, which is Split Tiny-ImageNet (Le & Yang, 2015) at 224×224 resolution. Finally, we test the performance of the s-OTDD with the OTDD as the reference in distance-driven data augmentation for image classification on CIFAR10 (Krizhevsky et al., 2009) dataset and Tiny-ImageNet (Le & Yang, 2015) dataset in Section 4.3. When dealing with the high dimensional datasets, i.e. CIFAR10 dataset and Tiny-ImageNet dataset, we use the convolution feature projection (Nguyen & Ho, 2022) for the s-OTDD while we use the Radon Transform projection for the NIST datasets and text feature datasets. We also provide experiments to compare the convolution feature projection and the Radon transform projection on the NIST datasets in Figure 12 in Appendix B. In general, the convolution-based projections require fewer projections yet exhibit stronger positive correlations between distance and adaptation performance. For all experiments, we use s-OTDD with $k = 5$ and $\sigma(\Lambda^k)$ be the product of $k$ Truncated Poisson distributions which have the corresponding

rate parameters be $1, \ldots, 5$ in turn.[1]

## 4.1. Empirical Analysis of Sliced Optimal Transport Dataset Distance

**Distance Correlation.** We treat the OTDD (Exact) as the desired dataset distance. After that, we compare the proposed s-OTDD with OTDD (Gaussian approximation), WTE, and CHSW in terms of correlations with the OTDD. We randomly split MNIST and CIFAR10 to create subdataset pairs, each ranging in size from 5,000 to 10,000. For each split, we plot the results of each method and OTDD (Exact), and report their correlations. The results are fully presented in Figure 9 and Figure 10 in Appendix B. The process is repeated 10 times and is summarized in Figure 1. From the figure, we observe that the s-OTDD (10,000 projections) has significant correlations with the OTDD (Exact) in both datasets. Moreover, s-OTDD shows comparable performance with OTDD (Gaussian approximation) and WTE while being computationally faster than them (see the next analysis).

**Computational Time.** We compare the computational time of the proposed s-OTDD with existing discussed dataset distances. We randomly split the MNIST dataset and the CIFAR10 dataset into two subdatasets which has the size varied from 1,000 to the full size of corresponding dataset. The total runtime includes all preprocessing steps, such as estimating the dataset's mean and covariance, as well as performing MDS for WTE and CHSW. We report the computational time in Figure 2. We observe that s-OTDD, with different numbers of projections, scales efficiently as the dataset size increases. Additionally, when moving from MNIST to CIFAR10 (which has a higher feature dimension), the computation time for s-OTDD does not increase significantly compared to OTDDs, WTE, and CHSW. In contrast to s-OTDD, other dataset distances struggle with relatively large datasets i.e., they are able to run for dataset sizes below 30,000 but then crash due to memory limitations (see Appendix C for the detail of the computational infrastructure). Thank to the computational benefits, s-OTDD successfully runs on large datasets, proving that it is more scalable than other existing dataset distances.

**Projection Analysis.** In addition to investigating the effect of varying the number of projections on computational time in Figure 2, we assess the dependency of s-OTDD to the number of projections. Specifically, we vary the number of projections from 1,000 to 50,000 and examine datasets of different sizes: 1,000, 5,000, 10,000, 15,000, and 20,000, computing the correlations with the configuration using 50,000 projections in each case. The results are reported in Figure 3. From the figure, we see that the

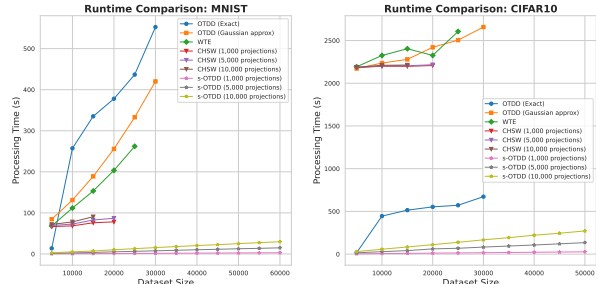

Figure 2: The figure shows computational time of OTDD (Exact), OTDD (Gaussian approx), WTE, CHSW (1,000, 5,000, 10,000 projections), and s-OTDD (1,000, 5,000, 10,000 projections) when varying size of two datasets.

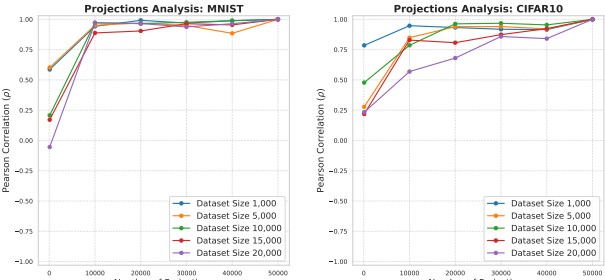

Figure 3: The figure shows Pearson correlations of s-OTDD with s-OTDD (50,000 projections) when varying number of projections from 1,000 to 50,000 in MNIST dataset and CIFAR10 dataset.

correlations grows very fast when increasing the number of projections, which strengthens the fast approximation rate in Proposition 3. Furthermore, to analyze how the number of projections influences the correlations with OTDD, we select a specific dataset size and compare the s-OTDD results using fewer projections (500, 1,000, 5,000, and 10,000) against the OTDD (Exact) baseline, with results presented in Figure 9 in Appendix B. The result indicates that increasing the number of projections enhances the alignment between s-OTDD and OTDD (Exact). We also conduct a similar projection anaylsis for CHSW in Figure 10 in Appendix B.

**Number of Moments Analysis.** The number of moments for the label projection is denoted by $k$ in the paper. It is noted that the higher $k$, the more moments information of the label distribution is gathered into the data point projection. Hence we want to have $k$ as big as possible as long as it does not raises numerical issue i.e., overflow which might be due to the non-existence of the higher moments. We conduct an ablation study in which $k$ is varied in Figure 7.

## 4.2. Transfer Learning

We apply the proposed method to transfer learning, following the OTDD framework. Transferability between source and target datasets is measured using the Performance Gap (PG), defined as the accuracy difference between fine-tuning on the full target dataset and using an adaptation method:

$$PG(\mathcal{D}_{S \to T}) = accuracy(\mathcal{D}_T) - accuracy(\mathcal{D}_{S \to T})$$

---

[1] All datasets and models were downloaded and evaluated at Movian AI or University of Texas at Austin

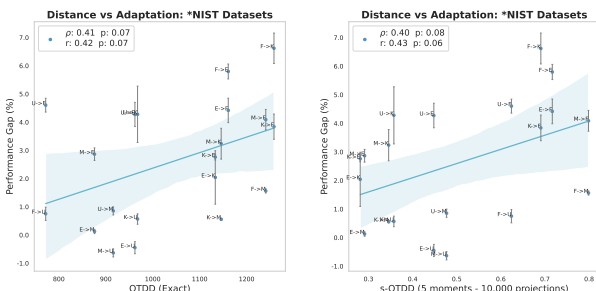

Figure 4: The figure shows correlations of OTDD (Exact) and s-OTDD (10,000 projections) with the performance gap when conducting transfer learning in *NIST datasets.

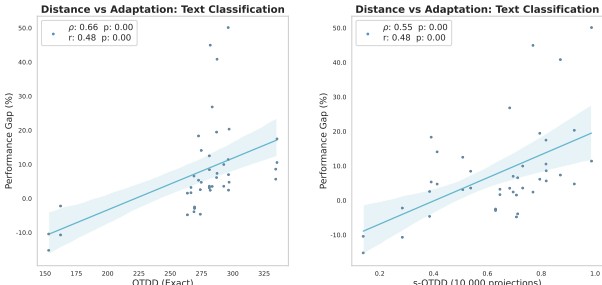

Figure 5: The figure shows correlations of OTDD (Exact) and s-OTDD (10,000 projections) with the performance when conducting transfer learning in text datasets.

PG serves a similar role to the Relative Drop metric in OTDD but offers a clearer view of adaptation transferability. A smaller PG indicates better adaptation for similar datasets, while a larger PG is expected for more distinct datasets.

**NIST Datasets.** We use a simplified LeNet-5, freezing the convolutional layers while fine-tuning the fully connected ones. We utilize the entire dataset for s-OTDD computation, while all other methods are evaluated using a random sample of 10,000 data points. Figure 4 shows a positive correlations between s-OTDD with 10,000 projections and PG ($\rho = 0.40$), which is roughly the same as OTDD (Exact). Additionally, OTDD (Gaussian approximation), CHSW, and WTE yield correlations of $\rho = 0.40$, $\rho = 0.16$, and $\rho = 0.37$, respectively (see Figure 11 in Appendix B).

**Text Datasets.** We limit the target dataset to 100 examples per class, fine-tune BERT on the source domain, then adapt and evaluate it on the target domain following OTDD settings. For distance calculations, we sample 5,000 points for OTDD (Exact) and 10,000 points for s-OTDD (10,000 projections). Sentences are embedded using BERT (base) (Li et al., 2022), and distances are computed on these embeddings. The results in Figure 5 shows that although OTDD (Exact) outweighs a bit higher than s-OTDD on Spearman rank, but both score the same on Pearson ($r = 0.48$). Crucially, s-OTDD performs the computation in roughly one-tenth the time, providing a substantial speed-efficiency advantage at the expense of a modest drop in Spearman.

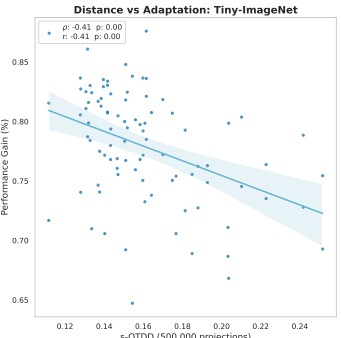

Figure 6: The figure shows the Pearson and Spearman correlations between s-OTDD (500,000 projections) and the performance gain in the Split Tiny-ImageNet (224×224) experiment.

**Split Tiny-ImageNet.** We randomly divide the Tiny ImageNet (Le & Yang, 2015) dataset into 10 disjoint tasks, each containing 20 classes. Each image is first rescaled to 256×256 and then center-cropped to obtain a 224×224 resolution. We initially train a ResNet-18 model on each task, serving as baseline, and subsequently freeze all layers except the final fully connected layer to fine-tune on the target task. We calculate the performance gain after transferbility by $Gain(\mathcal{D}_{S \to T}) = accuracy(\mathcal{D}_{S \to T}) - accuracy(\mathcal{D}_T)$. For distance calculations, we draw 5,000 sample pairs and compute s-OTDD with 500,000 random projections. As shown in Figure 6, s-OTDD exhibits a strong correlation with the performance gain. In contrast, OTDD is infeasible at the 224×224 resolution.

From above experiments, we observe that not only s-OTDD achieve comparable performance but also more scalable and robust to existing dataset distances in transfer learning while it offers considerably better computational speed.

### 4.3. Distance-Driven Data Augmentation

As discussed in (Alvarez-Melis & Fusi, 2020), data augmentation enhances dataset quality and diversity but lacks clear guidelines for its effective implementation. A model-agnostic dataset metric can help compare and select the most suitable augmentation strategies. Intuitively, a smaller distance between the augmented source dataset and the target dataset positively correlates with higher accuracy. In this large-scale experiment, we use augmented Tiny-ImageNet as the source dataset, CIFAR10 as the target dataset, and ResNet-50 (He et al., 2016) as the classifier. Augmentations for Tiny-ImageNet include random variations in brightness, contrast, saturation (0.1-0.9), and hue (0-0.5). We evaluate the proposed method and OTDD (Exact), with results presented in Figure 8. For s-OTDD, we sample 50,000 data points per dataset and use 100,000 projections, while for OTDD (Exact), we sample 5,000 data points per dataset. The results are summarized in Figure 8.

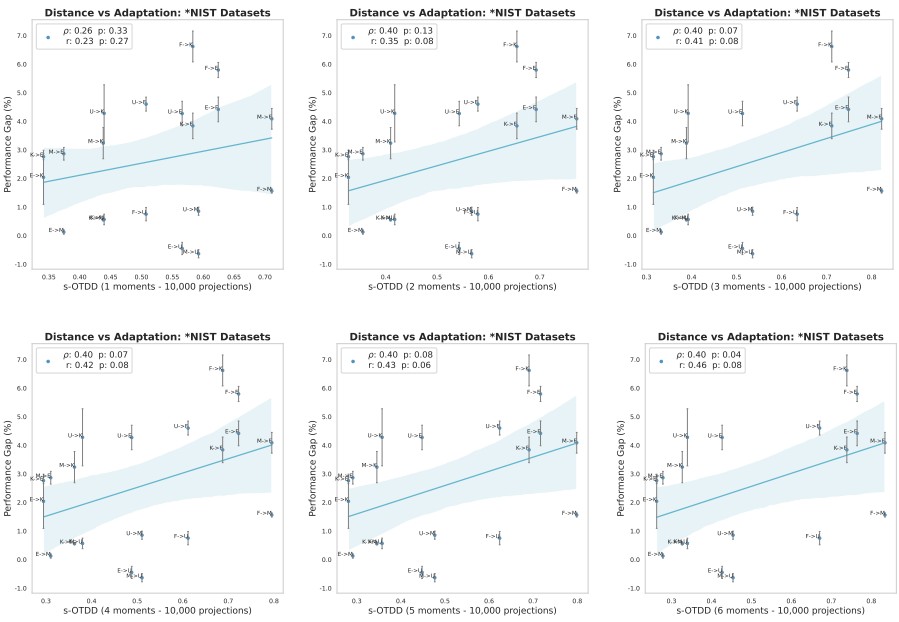

Figure 7: Experiment when varying number of moments in s-OTDD from 1 to 6 in *NIST Adaptation Experiment.

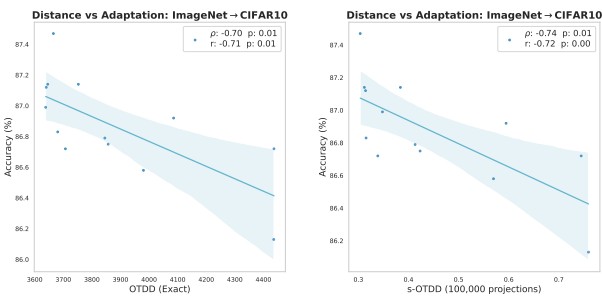

Figure 8: The figure shows correlations of OTDD (Exact) OTDD (Exact) and s-OTDD (100,000 projections) with the classification accuracy in data augmentation experiment.

Figure 8 shows that both OTDD (Exact) and our proposed s-OTDD method produce distances that are negatively correlated, which is natural, with test accuracy when transferring from augmented Tiny-ImageNet to CIFAR10. Despite processing ten times the dataset size compared to OTDD (Exact), s-OTDD is significantly faster, with a processing time in about $53 \times 10^3$ seconds, whereas OTDD (Exact) takes over $74 \times 10^3$ seconds, revealing a huge improvement in efficiency. Moreover, OTDD (Exact) achieves a Spearman correlations of $-0.70$, while s-OTDD attains a better correlations of $-0.74$. In other words, s-OTDD provides a reliable alternative distance to OTDD (Exact), achieving comparable performance while being faster.

## 5. Conclusion

We propose the *sliced optimal transport dataset distance* (s-OTDD), a versatile approach for comparing datasets that is model-agnostic, embedding-agnostic, training-free, and robust to variations in class count and disjoint label sets. At the heart of s-OTDD is the Moment Transform Projection (MTP), which encodes a label-represented as a feature distribution-into a real number. This projection enables us to represent entire datasets as one-dimensional distributions by mapping individual data points accordingly. From theoretical aspects, we discuss theoretical properties of the s-OTDD including metricity properties and approximation rate. From the computational aspects, the proposed s-OTDD achieves (near-)linear computational complexity with respect to the number of data points and feature dimensions, while remaining independent of the number of classes. For experiments, we analyze the performance of s-OTDD by comparing its computational time with existing dataset distances, its correlations with OTDD, its dependence on the number of projections, and number of moments, using subsets of the MNIST and CIFAR10 datasets. Moreover, we evaluate the correlations between s-OTDD and transfer learning performance gaps on image NIST datasets, text datasets and large scale Split Tiny-ImageNet (224x224 resolution). Finally, we test s-OTDD for data augmentation in image classification on CIFAR10 and Tiny-ImageNet. Overall, s-OTDD is comparable to baseline while being significantly faster, more scalable, and more robust. Future works will focus on understanding the gradient flow (Alvarez-Melis & Fusi, 2021) of the s-OTDD and adapt the s-OTDD in continual learning applications (Lee et al., 2021; Ke et al., 2020; Yang & Cai, 2023; Goldfarb et al., 2024).

## Impact Statement

This paper presents work to advance the field of Machine Learning. There are potential societal consequences of our work, none of which we feel must be highlighted.

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

# Supplement to "Lightspeed Geometric Dataset Distance via Sliced Optimal Transport"

We first give skipped proofs in the main text in Appendix A. We then present additional materials including algorithms and additional experiments in Appendix B. Finally, we report computational devices used for experiments in Appendix C.

## A. Proofs

### A.1. Proof of Proposition 1

We can rewrite the MTP in Definition 2 as:

$$\mathcal{MTP}_{\lambda,\theta}(\mu) = \frac{\int_{\mathbb{R}^d}(\mathcal{FP}_\theta(x))^\lambda f_\mu(x)dx}{\lambda!} = \frac{\int_{\mathbb{R}^d} t^\lambda f_{\mathcal{FP}_\theta\sharp\mu}(t)dt}{\lambda!},$$

where $f_{\mathcal{FP}_\theta\sharp\mu}$ is the density function of $\mathcal{FP}_\theta\sharp\mu$. Let $T_{\mu,\theta}$ be the random variable of $\mathcal{FP}_\theta\sharp\mu$, we have:

$$\mathcal{MTP}_{\lambda,\theta}(\mu) = \frac{\mathbb{E}[T_{\mu,\theta}^\lambda]}{\lambda!}.$$

For a given $\theta$, when $\mathcal{MTP}_{\lambda,\theta}(\mu) = \mathcal{MTP}_{\lambda,\theta}(\nu)$ for all $\lambda \in \Lambda$, it implies $\frac{\mathbb{E}[T_{\mu,\theta}^\lambda]}{\lambda!} = \frac{\mathbb{E}[T_{\nu,\theta}^\lambda]}{\lambda!}$ for all $\lambda \in \Lambda$.

(1) When $\Lambda$ is infinite i.e., $\Lambda = \mathbb{N}$, we have $\frac{\mathbb{E}[T_{\mu,\theta}^\lambda]}{\lambda!} = \frac{\mathbb{E}[T_{\nu,\theta}^\lambda]}{\lambda!}$ for all $\lambda \subset \mathbb{N}$. Therefore, we have:

$$\sum_{\lambda=1}^\infty \frac{z^\lambda \mathbb{E}[T_{\mu,\theta}^\lambda]}{\lambda!} = \sum_{\lambda=1}^\infty \frac{z^\lambda \mathbb{E}[T_{\nu,\theta}^\lambda]}{\lambda!}.$$

Since we assume that all projected scaled moment exists in Assumption 1 and the moment generating functions of $\mathcal{FP}_\theta\sharp\mu$ and $\mathcal{FP}_\theta\sharp\nu$ exist for all $\theta \in \Theta$, using Taylor's expansion, we have:

$$\mathbb{E}[e^{zT_{\mu,\theta}}] = \mathbb{E}[e^{zT_{\nu,\theta}}],$$

which means that moment generating function of $\mathcal{FP}_\theta\sharp\mu$ equals moment generating function of $\mathcal{FP}_\theta\sharp\nu$. Therefore, we can conclude that $\mathcal{FP}_\theta\sharp\mu = \mathcal{FP}_\theta\sharp\nu$ for all $\theta \in \Theta$ due to the uniqueness of the moment generating function. Due to the assumption of injectivity of the feature projection, we obtain $\mu = \nu$. We complete the proof.

(2) When $\Lambda$ is finite i.e., $\Lambda = \{1, 2, \ldots, \lambda_{max}\}$, we assume that projected Hankel matrices

$$A_{\theta,\mu} = \begin{bmatrix} m_{\theta,\mu,0} & m_{\theta,\mu,1} & \cdots & m_{\theta,\mu,\lambda_{\max}} \\ m_{\theta,\mu,1} & m_{\theta,\mu,2} & \cdots & m_{\theta,\mu,0} \\ \vdots & \vdots & \ddots & \vdots \\ m_{\theta,\mu,\lambda_{\max}} & m_{\theta,\mu,0} & \cdots & m_{\theta,\mu,\lambda_{\max}-1} \end{bmatrix}, A_{\theta,\nu} = \begin{bmatrix} m_{\theta,\nu,0} & m_{\theta,\nu,1} & \cdots & m_{\theta,\nu,\lambda_{\max}} \\ m_{\theta,\nu,1} & m_{\theta,\nu,2} & \cdots & m_{\theta,\nu,0} \\ \vdots & \vdots & \ddots & \vdots \\ m_{\theta,\nu,\lambda_{\max}} & m_{\theta,\nu,0} & \cdots & m_{\theta,\nu,\lambda_{\max}-1} \end{bmatrix}$$

, with $m_{\theta,\mu,\lambda} = \int_{\mathbb{R}^d}(\mathcal{FP}_\theta(x))^\lambda f_\mu(x)dx$ and $m_{\theta,\nu,\lambda} = \int_{\mathbb{R}^d}(\mathcal{FP}_\theta(x))^\lambda f_\nu(x)dx$, are positive definite for all $\theta \in \Theta$ and $|m_{\theta,\mu,\lambda}| < CD^\lambda \lambda!$, and $|m_{\theta,\nu,\lambda}| < CD^\lambda \lambda!$ for some constants $C$ and $D$ for all $\lambda \in \Lambda$. From the Hamburger moment problem (Reed & Simon, 1975; Chihara, 2011), we know that moments uniquely define a distribution. Therefore, we have $\mathcal{FP}_\theta\sharp\mu = \mathcal{FP}_\theta\sharp\nu$ for all $\theta \in \Theta$. Due to the assumption of injectivity of the feature projection, we obtain $\mu = \nu$. We complete the proof.

### A.2. Proof of Proposition 2

From Definition 4, we have:

$$\text{s-OTDD}_p^p(\mathcal{D}_1, \mathcal{D}_2) = \mathbb{E}_{(\psi,\theta,\lambda,\phi)\sim\mathcal{U}(\mathbb{S})\otimes\mathcal{U}(\mathbb{S}^{d-1})\otimes\sigma(\Lambda^k)\otimes\mathcal{U}(\Phi)}\left[\text{W}_p^p(\mathcal{DP}_{\psi,\theta,\lambda,\phi}^k\sharp P_{\mathcal{D}_1}, \mathcal{DP}_{\psi,\theta,\lambda,\phi}^k\sharp P_{\mathcal{D}_2})\right].$$

Since the Wasserstein distance is non-negative and symmetric (Peyré & Cuturi, 2020), the symmetry and non-negativity of the s-OTDD$_p^p(\mathcal{D}_1, \mathcal{D}_2)$ follows directly from it. We now prove the triangle inequality of s-OTDD. Given any three datasets $\mathcal{D}_1, \mathcal{D}_2$, and $\mathcal{D}_3$. We want to show that:

$$\text{s-OTDD}_p(\mathcal{D}_1, \mathcal{D}_2) \leq \text{s-OTDD}_p(\mathcal{D}_1, \mathcal{D}_3) + \text{s-OTDD}_p(\mathcal{D}_3, \mathcal{D}_2).$$

From the triangle inequality of the Wasserstein distance, we have:

$$\text{s-OTDD}_p(\mathcal{D}_1, \mathcal{D}_2) = \left( \mathbb{E}_{(\psi,\theta,\lambda,\phi)\sim\mathcal{U}(\mathbb{S})\otimes\mathcal{U}(\mathbb{S}^{d-1})\otimes\sigma(\Lambda^k)\otimes\mathcal{U}(\Phi)} \left[ \text{W}_p^p(\mathcal{DP}_{\psi,\theta,\lambda,\phi}^k\sharp P_{\mathcal{D}_1}, \mathcal{DP}_{\psi,\theta,\lambda,\phi}^k\sharp P_{\mathcal{D}_2}) \right] \right)^{\frac{1}{p}}$$

$$\leq \left( \mathbb{E}_{(\psi,\theta,\lambda,\phi)\sim\mathcal{U}(\mathbb{S})\otimes\mathcal{U}(\mathbb{S}^{d-1})\otimes\sigma(\Lambda^k)\otimes\mathcal{U}(\Phi)} \left[ (\text{W}_p(\mathcal{DP}_{\psi,\theta,\lambda,\phi}^k\sharp P_{\mathcal{D}_1}, \mathcal{DP}_{\psi,\theta,\lambda,\phi}^k\sharp P_{\mathcal{D}_3}) \right.\right.$$

$$\left.\left. + \text{W}_p(\mathcal{DP}_{\psi,\theta,\lambda,\phi}^k\sharp P_{\mathcal{D}_3}, \mathcal{DP}_{\psi,\theta,\lambda,\phi}^k\sharp P_{\mathcal{D}_2}))^p \right] \right)^{\frac{1}{p}}.$$

Using the Minkowski's inequality, we further have:

$$\text{s-OTDD}_p(\mathcal{D}_1, \mathcal{D}_2) \leq \left( \mathbb{E}_{(\psi,\theta,\lambda,\phi)\sim\mathcal{U}(\mathbb{S})\otimes\mathcal{U}(\mathbb{S}^{d-1})\otimes\sigma(\Lambda^k)\otimes\mathcal{U}(\Phi)} \left[ \text{W}_p^p(\mathcal{DP}_{\psi,\theta,\lambda,\phi}^k\sharp P_{\mathcal{D}_3}, \mathcal{DP}_{\psi,\theta,\lambda,\phi}^k\sharp P_{\mathcal{D}_2}) \right] \right)^{\frac{1}{p}}$$

$$+ \left( \mathbb{E}_{(\psi,\theta,\lambda,\phi)\sim\mathcal{U}(\mathbb{S})\otimes\mathcal{U}(\mathbb{S}^{d-1})\otimes\sigma(\Lambda^k)\otimes\mathcal{U}(\Phi)} \left[ \text{W}_p^p(\mathcal{DP}_{\psi,\theta,\lambda,\phi}^k\sharp P_{\mathcal{D}_3}, \mathcal{DP}_{\psi,\theta,\lambda,\phi}^k\sharp P_{\mathcal{D}_2}) \right] \right)^{\frac{1}{p}}$$

$$= \text{s-OTDD}_p(\mathcal{D}_1, \mathcal{D}_3) + \text{s-OTDD}_p(\mathcal{D}_3, \mathcal{D}_2),$$

which completes the proof of the triangle inequality. For the identity of indiscernibles, when $\mathcal{D}_1 = \mathcal{D}_2$, we have directly that

$$\text{s-OTDD}_p(\mathcal{D}_1, \mathcal{D}_2) = \left( \mathbb{E}_{(\psi,\theta,\lambda,\phi)\sim\mathcal{U}(\mathbb{S})\otimes\mathcal{U}(\mathbb{S}^{d-1})\otimes\sigma(\Lambda^k)\otimes\mathcal{U}(\Phi)} \left[ \text{W}_p^p(\mathcal{DP}_{\psi,\theta,\lambda,\phi}^k\sharp P_{\mathcal{D}_1}, \mathcal{DP}_{\psi,\theta,\lambda,\phi}^k\sharp P_{\mathcal{D}_2}) \right] \right)^{\frac{1}{p}}$$

$$= \left( \mathbb{E}_{(\psi,\theta,\lambda,\phi)\sim\mathcal{U}(\mathbb{S})\otimes\mathcal{U}(\mathbb{S}^{d-1})\otimes\sigma(\Lambda^k)\otimes\mathcal{U}(\Phi)} [0] \right)^{\frac{1}{p}} = 0,$$

due to the identity of indiscernibles of the Wasserstein distance when $\mathcal{DP}_{\psi,\theta,\lambda,\phi}^k\sharp P_{\mathcal{D}_1} = \mathcal{DP}_{\psi,\theta,\lambda,\phi}^k\sharp P_{\mathcal{D}_2}$. In addition, when $\text{s-OTDD}_p(\mathcal{D}_1, \mathcal{D}_2) = 0$, it implies that $\text{W}_p^p(\mathcal{DP}_{\psi,\theta,\lambda,\phi}^k\sharp P_{\mathcal{D}_1}, \mathcal{DP}_{\psi,\theta,\lambda,\phi}^k\sharp P_{\mathcal{D}_2}) = 0$ for all $\psi \in \mathbb{S}, \theta \in \Theta, \lambda \in \Lambda, \phi \in \Phi$. As a result, $\mathcal{DP}_{\psi,\theta,\lambda,\phi}^k\sharp P_{\mathcal{D}_1} = \mathcal{DP}_{\psi,\theta,\lambda,\phi}^k\sharp P_{\mathcal{D}_2}$ for all $\psi \in \mathbb{S}, \theta \in \Theta, \lambda \in \Lambda, \phi \in \Phi$. From the assumption of the injectivity of the data point projection, we obtain that $P_{\mathcal{D}_1} = P_{\mathcal{D}_2}$ which implies $\mathcal{D}_1 = \mathcal{D}_2$. We conlude the proof here.

### A.3. Proof of Proposition 3

We recall the empirical approximation of the s-OTDD is:

$$\widehat{\text{s-OTDD}}_p^p(\mathcal{D}_1, \mathcal{D}_2; L) = \frac{1}{L}\sum_{l=1}^{L} \text{W}_p^p(\mathcal{DP}_{\psi_l,\theta_l,\lambda_l,\phi_l}^k\sharp P_{\mathcal{D}_1}, \mathcal{DP}_{\psi_l,\theta_l,\lambda_l,\phi_l}^k\sharp P_{\mathcal{D}_2}).$$

Using Holder's inequality, we have:

$$\mathbb{E}\left[ \left| \widehat{\text{s-OTDD}}_p^p(\mathcal{D}_1, \mathcal{D}_2; L) - \text{s-OTDD}_p^p(\mathcal{D}_1, \mathcal{D}_2) \right| \right]$$

$$\leq \left( \mathbb{E}\left[ \left| \widehat{\text{s-OTDD}}_p^p(\mathcal{D}_1, \mathcal{D}_2; L) - \text{s-OTDD}_p^p(\mathcal{D}_1, \mathcal{D}_2) \right|^2 \right] \right)^{\frac{1}{2}}$$

$$= \left( \mathbb{E}\left[ \left| \frac{1}{L}\sum_{l=1}^{L} \text{W}_p^p(\mathcal{DP}_{\psi_l,\theta_l,\lambda_l,\phi_l}^k\sharp P_{\mathcal{D}_1}, \mathcal{DP}_{\psi_l,\theta_l,\lambda_l,\phi_l}^k\sharp P_{\mathcal{D}_2}) - \mathbb{E}\left[ \text{W}_p^p(\mathcal{DP}_{\psi,\theta,\lambda,\phi}^k\sharp P_{\mathcal{D}_1}, \mathcal{DP}_{\psi,\theta,\lambda,\phi}^k\sharp P_{\mathcal{D}_2}) \right] \right|^2 \right] \right)^{\frac{1}{2}}.$$

Since

$$\mathbb{E}\left[ \frac{1}{L}\sum_{l=1}^{L} \text{W}_p^p(\mathcal{DP}_{\psi_l,\theta_l,\lambda_l,\phi_l}^k\sharp P_{\mathcal{D}_1}, \mathcal{DP}_{\psi_l,\theta_l,\lambda_l,\phi_l}^k\sharp P_{\mathcal{D}_2}) \right] = \frac{1}{L}\sum_{l=1}^{L} \mathbb{E}[\text{W}_p^p(\mathcal{DP}_{\psi_l,\theta_l,\lambda_l,\phi_l}^k\sharp P_{\mathcal{D}_1}, \mathcal{DP}_{\psi_l,\theta_l,\lambda_l,\phi_l}^k\sharp P_{\mathcal{D}_2})]$$

$$= \mathbb{E}\left[ \text{W}_p^p(\mathcal{DP}_{\psi,\theta,\lambda,\phi}^k\sharp P_{\mathcal{D}_1}, \mathcal{DP}_{\psi,\theta,\lambda,\phi}^k\sharp P_{\mathcal{D}_2}) \right],$$

---

**Algorithm 1** Computational Algorithm for s-OTDD

---

**Input:** Two datasets $\mathcal{D}_1 = \{(x_1, y_1), \ldots, (x_n, y_n)\}$ and $\mathcal{D}_2 = \{(x'_1, y'_1), \ldots, (x'_m, y'_m)\}$, label sets $\mathcal{Y}_1$ and $\mathcal{Y}_2$, parameter $p \geq 1$, number of moments $k$, number of projections $L$, rate parameters $r_1, \ldots, r_k > 0$.

**for** $l = 1$ to $L$ **do**
   Sample $\theta_l \sim \mathcal{U}(\mathbb{S}^{d-1})$
   **for** $i = 1$ to $n$ **do**
      Compute $\mathcal{FP}_\theta(x_i)$
   **end for**
   **for** $j = 1$ to $m$ **do**
      Compute $\mathcal{FP}_\theta(x'_j)$
   **end for**
   **for** $j = 1$ to $k$ **do**
      Sample $\lambda_l^{(j)} \sim \text{TruncatedPoisson}(r_j)$
      **for** $a$ in $\mathcal{Y}_1$ **do**
         Compute $\mathcal{MTP}_{\lambda^{(i)},\theta}(q_a) = \frac{1}{n_a} \sum_{i=1}^{n_a} \frac{\mathcal{FP}_\theta(x_i)^{\lambda^{(j)}}}{\lambda!}$ for $n_a$ is the number of samples in class $a$.
      **end for**
      **for** $a'$ in $\mathcal{Y}_2$ **do**
         Compute $\mathcal{MTP}_{\lambda^{(i)},\theta}(q_{a'}) = \frac{1}{m_{a'}} \sum_{i=1}^{m_{s'}} \frac{\mathcal{FP}_\theta(x'_i)^{\lambda^{(j)}}}{\lambda!}$ for $m_{a'}$ is the number of samples in class $a'$.
      **end for**
   **end for**
   Sample $\psi_l \sim \mathcal{U}(\mathbb{S}^k)$
   **for** $i = 1$ to $n$ **do**
      $\mathcal{DP}^k_{\psi,\theta,\lambda,\theta}(x_i, q_{y_i}) = \psi^{(1)}\mathcal{FP}_\theta(x_i) + \sum_{i'=1}^k \psi^{(k)}\mathcal{MTP}_{\lambda^{(i')},\theta}(q_{y_i})$
   **end for**
   **for** $i = j$ to $m$ **do**
      $\mathcal{DP}^k_{\psi,\theta,\lambda,\theta}(x'_j, q_{y'_j}) = \psi^{(1)}\mathcal{FP}_\theta(x'_j) + \sum_{i'=1}^k \psi^{(k)}\mathcal{MTP}_{\lambda^{(i')},\theta}(q_{y'_j})$
   **end for**
   Compute $w_l = W_p^p \left( \frac{1}{n} \sum_{i=1}^n \delta_{\mathcal{DP}^k_{\psi,\theta,\lambda,\theta}(x_i,q_{y_i})}, \frac{1}{m} \sum_{j=1}^m \delta_{\mathcal{DP}^k_{\psi,\theta,\lambda,\theta}(x'_j,q_{y'_j})} \right)$
**end for**
**Return:** $\frac{1}{L} \sum_{l=1}^l w_l$

---

we have:

$$\mathbb{E}\left[\left|\widehat{\text{s-OTDD}}_p^p(\mathcal{D}_1, \mathcal{D}_2; L) - \text{s-OTDD}_p^p(\mathcal{D}_1, \mathcal{D}_2)\right|\right]$$

$$\leq \left( Var\left[ \frac{1}{L} \sum_{l=1}^L \mathrm{W}_p^p(\mathcal{DP}^k_{\psi_l,\theta_l,\lambda_l,\phi_l} \sharp P_{\mathcal{D}_1}, \mathcal{DP}^k_{\psi_l,\theta_l,\lambda_l,\phi_l} \sharp P_{\mathcal{D}_2}) \right] \right)^{\frac{1}{2}}$$

$$= \frac{1}{\sqrt{L}} \left( Var\left[ \mathrm{W}_p^p(\mathcal{DP}^k_{\psi,\theta,\lambda,\phi} \sharp P_{\mathcal{D}_1}, \mathcal{DP}^k_{\psi,\theta,\lambda,\phi} \sharp P_{\mathcal{D}_2}) \right] \right)^{\frac{1}{2}},$$

due to the i.i.d sampling of the projection parameters, which completes the proof.

## B. Additional Materials

**Algorithms.** We present the computational algorithm for s-OTDD in Algorithm 1. In the algorithm, we use the same feature projection for both the label projection (MTP) and the data point projection.

**Projection Analysis.** As mentioned in the main text, we present the distance correlations with OTDD (Exact) of s-OTDD as a function of the number of projections in Figure 9 and similarly for CHSW in Figure 10. For s-OTDD, we observe that increasing the number of projections consistently improves the correlations for both MNIST and CIFAR10 datasets, indicating a clear trend. In contrast, the same does not hold true for CHSW, where the correlations do not show consistent

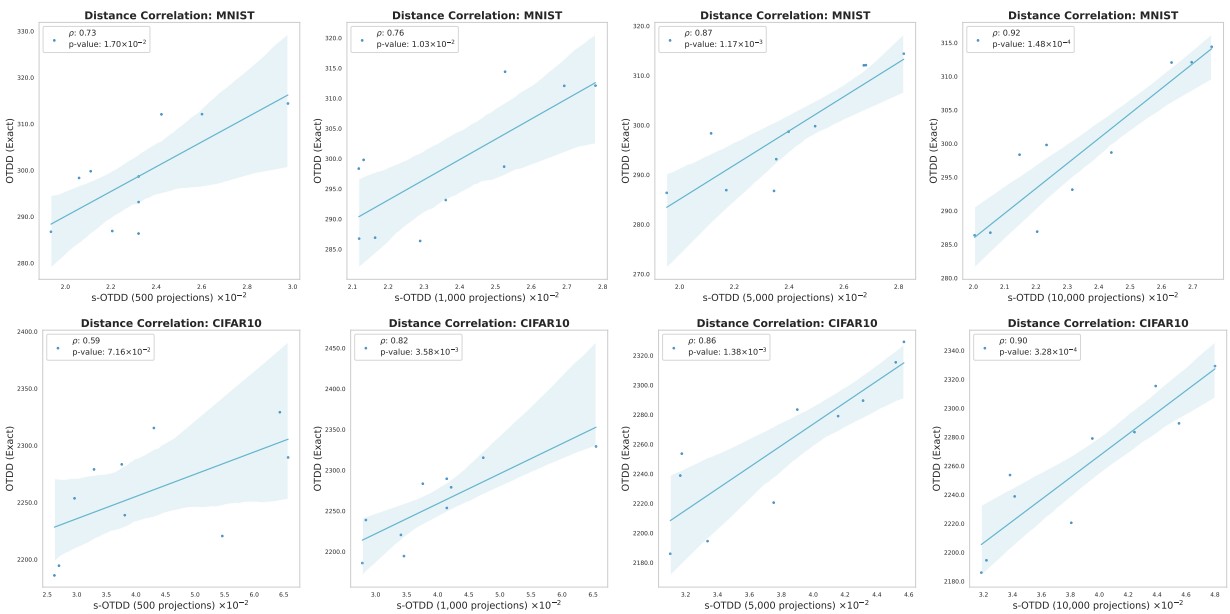

Figure 9: The figure shows distance correlations with OTDD (Exact) of s-OTDD when varying the number of projections.

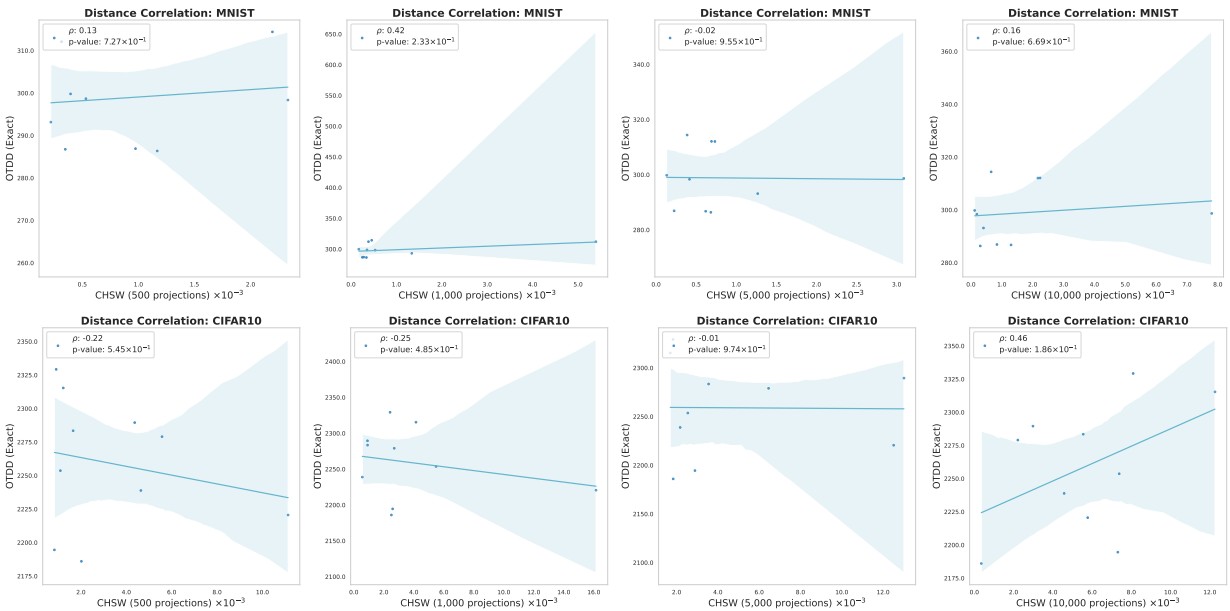

Figure 10: The figure shows distance correlation with OTDD (Exact) of CHSW when varying the number of projections.

improvement as the number of projections increases. This important observation suggests that s-OTDD in its population form is highly correlated with OTDD (Exact), and therefore, reducing the Monte Carlo approximation error by increasing the number of projections results in a more accurate correlations with OTDD (Exact).

**Transfer Learning.** We present the complete experimental results for transfer learning using the *NIST datasets in Figure 11. As highlighted in Section 4.2, the figure demonstrates clear positive correlations between s-OTDD with 10,000 projections and PG, with a Spearman's rank correlation coefficient of $\rho = 0.42$, which is identical to that of OTDD (Exact). For the other distances, we observe that OTDD (Gaussian approximation), CHSW, and WTE yield correlations of $\rho = 0.44$, $\rho = 0.15$, and $\rho = 0.43$, respectively. These results reinforce the reliability of s-OTDD in capturing meaningful transfer learning behavior, comparable to OTDD (Exact) and other related metrics.

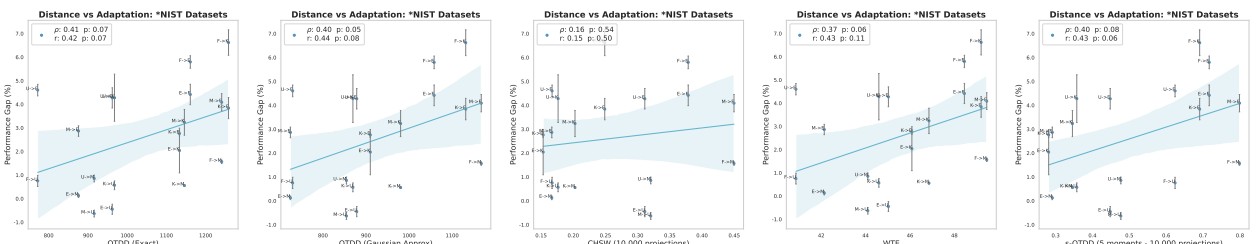

Figure 11: The figures show Pearson and Spearman correlation of OTDD (Exact), OTDD (Gaussian approximation), CHSW (10,000 projections), WTE, and s-OTDD (10,000 projections) with the performance gap when conducting transfer learning in *NIST datasets.

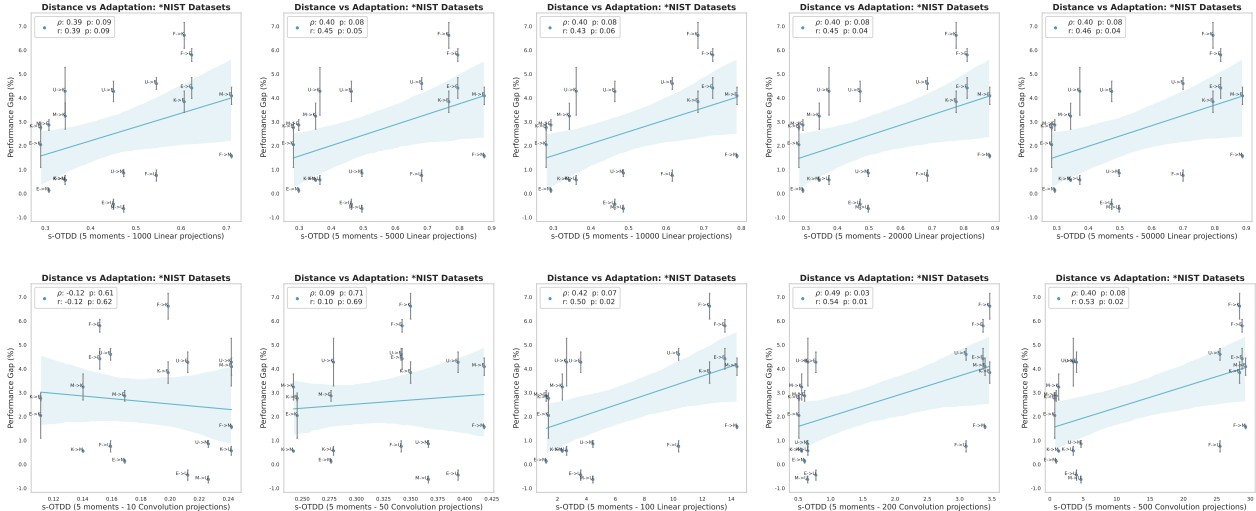

Figure 12: The figure compares two projection methods, linear projections (Radon Transform projection) and convolution projection, using the *NIST dataset experiments.

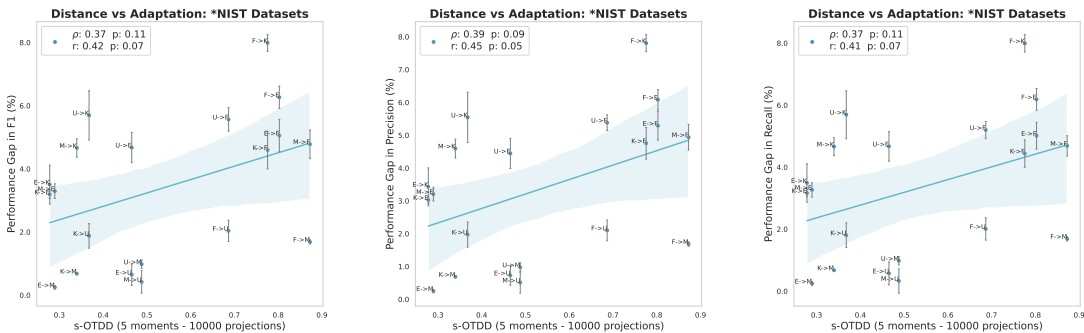

Figure 13: The figure shows Pearson and Spearman correlations of distance and Performance Gap in term of other metrics, i.e F1, Precision and Recall.

## C. Computational Devices

For the runtime experiments and distance computations, we conducted tests using 8 CPU cores with 128GB of memory. For model training experiments, such as training BERT and ResNet, we used an NVIDIA A100 GPU.

