# OpenReview forum: "Lightspeed Geometric Dataset Distance via Sliced Optimal Transport"
_ICML.cc/2025/Conference — ICML 2025 poster_

### Official Review · Reviewer_LbD9 · 2025-03-11

**Overall Recommendation:** 3

**Summary:**

The paper proposes a sliced optimal transport-based dataset distance measure that uses moment transformation projection. The method improves upon a previously proposed optimal transport based dataset distance measure and makes the dataset distance computation more efficient. The paper presents a theoretical analysis of this distance and discusses various properties of the distance measure along with the computational complexity. The paper then presents evaluation of this distance on tasks such as transfer learning and data augmentation.

## Update after the rebuttal
I thank the authors for providing detailed answers to my questions. I have updated my rating of the work (2 --> 3).

**Claims And Evidence:**

The theoretical claims are well supported and convincing while the empirical evaluation seems rather limited.

**Essential References Not Discussed:**

NA

**Experimental Designs Or Analyses:**

The applications of transfer learning and data augmentation are relevant.

**Methods And Evaluation Criteria:**

The evaluation methodology makes sense but are rather small scale making it unclear of whether the approach is effective practically or not.

**Other Comments Or Suggestions:**

1. A large-scale evaluation on high dimensional datasets is required to demonstrate the effectiveness of the method over OTDD.
2. Answers to below questions will be helpful for understanding the utility of the approach beyond just being faster than OTDD.

**Other Strengths And Weaknesses:**

Strengths:
1. The problem of measuring dataset distance is very relevant.
2. The proposal to use OT-based method and improving the efficiency over existing OTDD work is a good contribution.
3. Theoretical analysis of the proposed metric is sound.

Weaknesses:
1. The empirical evaluation of the method is very basic and follows the OTDD paper very closely with no new experiments or datasets to demonstrate the effectiveness of s-OTDD beyond what had already been shown. Since efficiency is the main contribution, the work should include at least a large-scale experiment on an application where OTDD is not applicable/ very expensive and s-OTDD handles it with ease.
2. The datasets used in the work to compute the distance tend to be very simplistic such as *NIST-type datasets.
3. The reason to use Pearson Correlation between OTDD and s-OTDD for evaluation is not clear.

**Questions For Authors:**

1. Why is OTDD a natural choice for dataset distance? Why should a new dataset distance try to be correlated with OTDD? Why should the correlation be the Pearson correlation (wouldn't a rank correlation be a better way of measuring the effectiveness of the new metric)?

2. The number of random projections that lead to the best result seem to be of the same order as the number of samples in the dataset. What are the implications of it for larger datasets? The time complexity for the method increases to $O(n^2logn)$  and space complexity $O(n(n+d))$. Can you comment about this and give a guideline of how many projections should one be looking at for the best results?

3. The results of sec 4.2 and 4.3 have shown that s-OTDD achieves similar correlation to accuracy for transfer learning/data augmentation problems compared to OTDD. But since s-OTDD can work with more data that OTDD, why is it not more predictive of the transfer learning/data augmentation performance?

4. Is s-OTDD predictive of metrics other than classification accuracy?

5. What types of datasets/tasks (beyond single label classification datasets) can s-OTDD be used to measure distances for?

**Relation To Broader Scientific Literature:**

The paper builds on the paper titled "Geometric dataset distances via OT (OTDD)" which had initially proposed a measure to calculate dataset distances via OT. The paper proposes a sliced OT based method which improves the efficiency of the previous approach. The paper follows the same experimental protocol used in the OTDD paper and shows that their distance measure is correlated with OTDD while being more efficient.

**Theoretical Claims:**

The claims and proofs look good but I did not check the details of the proofs though.

---

> ### Author Rebuttal · Authors · 2025-03-31
>
> We would like to thank the reviewer for the time and constructive feedback.
>
> **Q11**. Please provide large-scale experiments where OTDD is too expensive, and s-OTDD excels.
>
> **A11.** In the paper, we showed that OTDD cannot be used when the dataset size reaches more than 30,000 samples on MNIST and CIFAR10 in Figure 2. In addition, we added new experiments on the large-scale Tiny-ImageNet dataset, where the OTDD baseline is not feasible to compute in **A14**.
>
> **Q12.** The datasets used in the work to compute the distance tend to be very simplistic, such as *NIST-type datasets*.
>
> **A12.** We did conduct text classification with various datasets such as AG News, DBPedia, Yelp Reviews, Amazon Reviews, and Yahoo Answers (Section 4.2). In addition, we conducted experiment on Tiny ImageNet and CIFAR-10 (32×32 resolution). We also refer the reviewer to  **A14** where we added a new experiment on Tiny-ImageNet of size 224×224.
>
> **Q13.** The reason to use Pearson Correlation between OTDD and s-OTDD for evaluation is not clear.
>
> **A13.** s-OTDD can be seen as an alternative solution for OTDD when dealing with a large scale datasets. Therefore, measuring correlation with OTDD strengthens the claim that s-OTDD can replace OTDD while being more efficient.
>
> **Q14.** A large-scale evaluation on high dimensional datasets is required to demonstrate the effectiveness of the method over OTDD.
>
> **A14.** We have added a large-scale transfer learning experiment in Tiny-ImageNet dataset (224x224 resolution) where computing OTDD is infeasible. To compute dataset distances, we sampled 5,000 examples from each sub-dataset. The result of the experiment is visible in Figure 16 at this link [https://imgur.com/a/hHvg2T2](https://imgur.com/a/hHvg2T2). In the figure, we see that s-OTDD has a relatively good correlation with the classification accuracy.
>
> **Q15.** Why is OTDD a natural choice for dataset distance? Why should a new dataset distance try to be correlated with OTDD? Why should the correlation be the Pearson correlation (wouldn't a rank correlation ...)?
>
> **A15.** OTDD is a good dataset distance since it is model-agnostic, does not involve training, can compare datasets even if their label sets are completely disjoint.  Similar to OTDD, s-OTDD is also model-agnostic (requires no modeling on data), does not need to estimate any parameters, and can handle label sets that are completely disjoint. Since s-OTDD is proposed as an alternative solution to OTDD, it needs to have a good correlation with OTDD.
>
> From the suggestion of the reviewer, we have added the new figures with both Pearson correlation (denoted as $r$) and Spearman's rank correlation (denoted as $\rho$) in Figure 12-14 at this link [https://imgur.com/a/hHvg2T2](https://imgur.com/a/hHvg2T2). We observe that the values of Spearman's rank correlation and Pearson correlation are almost similar in majority of cases for s-OTDD.
>
> **Q16**. The number of random projections ... What are the implications of it for larger datasets? The time complexity for the method increases to $O(n^2 \log n)$ and space complexity $O(n(n + d))$... how many projections should one be looking at for the best results?
>
> **A16.** In practice, $L$ should be large enough with respect to the number of dimensions $d$ not the dataset size $n$. In particular, authors in [4] empirically show that we need $L\approx 1.22 \sqrt{d}$. In Figure 2, we showed that the computation gap between $L=1000$ and $L=10000$ is not large. Moreover, the computations for $L$ projections are independent, hence, we can utilize parallel computing to compute s-OTDD. Therefore, having a large $L$ is not a problem. In addition, we can reduce $L$ by using more advanced sampling techniques [5].
>
> [4] Sliced Wasserstein Autoencoders, Kolouri et al.
> [5] Quasi-Monte Carlo for 3D Sliced Wasserstein, Nguyen et al.
>
> **Q17**. ... s-OTDD can work with more data than OTDD, why is it not more predictive ... ?
>
> **A17.**.  In the paper, we actually compare s-OTDD and OTDD with the same number of data samples. We want to convey that s-OTDD is much faster than OTDD while being comparable in performance.  In Figure 2 and in **A14**, we showed that OTDD cannot be used when having large or high-dimensional datasets.
>
> **Q18.** Is s-OTDD predictive of metrics other than classification accuracy?
>
> **A18.**  We can compute the correlation between s-OTDD and any function of two datasets.  We added a new experiments to show that s-OTDD also correlates well with other performance metrics such as Precision, Recall, and F1 Score in Figure 15  [https://imgur.com/a/hHvg2T2](https://imgur.com/a/hHvg2T2).
>
> **Q19.** What types of datasets/tasks (...) can s-OTDD be used to measure distances for?
>
> **A19.** s-OTDD can also be interpreted as a distance between distributions over distributions. Therefore, s-OTDD can also be adapted to compare  distributions over point clouds, distributions over histograms, distributions over 3D shapes, multi-label datasets, and so on.

---

> > ### Comment · Reviewer_LbD9 · 2025-04-03
> >
> > I thank the authors for their responses to my questions. I have some follow up questions
> >
> > 1. Authors mention that OTDD cannot be computed on more that 30,000 samples in A11 but for the experiment in A14 where they demonstrate the advantage of s-OTDD uses a subset of 5000 samples. Why was a subset necessary if s-OTDD can handle all the samples? What happens to the correlation if more samples are used?
> >
> > 2. What do the points on Fig 16 refer to? What is the computational time (wall clock time) required to compute s-OTDD on this dataset. How does it compare to *NIST datasets in Sec 4.2 vs OTDD.
> >
> > 3. In A17, does adding more data improve the correlation of s-OTDD to the metrics?

---

> > > ### Author Response · Authors · 2025-04-04
> > >
> > > We would like to thank the reviewer for the additional questions. We would like to answer them as follows:
> > >
> > > **Q1.** Authors mention that OTDD cannot be computed on more that 30,000 samples in A11 but for the experiment in A14 where they demonstrate the advantage of s-OTDD uses a subset of 5000 samples. Why was a subset necessary if s-OTDD can handle all the samples? What happens to the correlation if more samples are used?
> > >
> > > **A1.**  It is worth noting that the computational time and memory also depend on the number of dimensions, therefore, OTDD  cannot be computed for this case of 5000 samples (in a high-dimension 3x224x224). To calculate the correlation, we need a random generating distribution to generate datasets. Therefore,  we use bootstrap sampling to obtain such generating distribution from a given fixed dataset. From the bootstrap distribution, we create random subsets (smaller datasets) to calculate multiple pairs of distance and the function of interest of two datasets with which we want to form the correlation.  The reason we used 5000 samples (in a high-dimension 3x224x224)  was to keep the computational time reasonable, given the time constraint that we had during the rebuttal. By increasing the bootstrap size,  the variance of estimations e.g., distances, functions of interests,  and correlation, will reduce due to the reduction of variability between random subsets (variation of the bootstrap distribution).   We refer the reader to Figure 9 in the original OTDD paper [1] for an empirical study of this behaviour.  By increasing the bootstrap size, the estimation of correlation and other functions of two datasets will converge to a population value.
> > >
> > > [1] Geometric Dataset Distances via Optimal Transport, Alvarez-Melis et al.
> > >
> > >
> > > **Q2.** What do the points on Fig 16 refer to? What is the computational time (wall clock time) required to compute s-OTDD on this dataset. How does it compare to *NIST datasets in Sec 4.2 vs OTDD.
> > >
> > > **A2.** In Figure 16, the x-axis represents values of s-OTDD distance, and the y-value represents the value of the accuracy gain. For each random subset, we have one point in the figure. For this setup with 5000 samples of size 3x224x224, computing s-OTDD costs about 3450.73 (s) compared to about 40 (s) for MNIST with 60000 samples of size 28x28.  For OTDD, we cannot compute it in this high-dimensional setting due to an out-of-memory problem. We refer the reviewer to Figure 2 for a relative comparison of computational time when varying dataset size on MNIST.
> > >
> > > **Q3.** In A17, does adding more data improve the correlation of s-OTDD to the metrics?
> > >
> > > **A3.**  We recall that correlation can only be used to compare dataset distances with the same bootstrap size (random subset size). Varying the bootstrap size changes the dataset-generating distributions, which in turn alters the distributions of dataset distances and functions of interest. As a result, the corresponding correlations belong to different dataset distributions and are not directly comparable. As discussed, we only know that increasing the bootstrap size causes the correlation to converge to some population value. Therefore, the notion of "improvement" when adding more data is not well-defined.
> > >
> > > The correlation computation in the paper is intended solely for relative comparison among s-OTDD, OTDD, CHSW, and WTE with datasets of the **same size**. In practice, computing correlation and creating sub-datasets is unnecessary. Our key claim is that s-OTDD is scalable and can replace OTDD, as it exhibits similar correlations to OTDD and other metrics when the dataset-generating distributions (obtained via bootstrapping) remain the same.
> > >
> > >
> > > We would like to thank the reviewer again for letting us explain more about the details. Please feel free to let us know if you still have any concerns.
> > >
> > > Best regards,
> > >
> > > Authors,

---

### Official Review · Reviewer_FRUy · 2025-03-14

**Overall Recommendation:** 4

**Summary:**

The authors propose a similarity measure for supervised tasks based on sliced optimal transport. The data with labels are mapped to 1D slices via a feature projection map for features and the induced moment transform projection for label distributions. Then the dataset distance is defined using the 1D optimal transport distances. The proposed distance is shown to be a proper metric. Numerically, it is effective and efficient in predicting task transferability over multiple image and text datasets.

**Claims And Evidence:**

1. Proposition 1 gives two conditions for the MTP to be injective, and the metric properties of s-OTDD are based on the injectivity of MTP. Does the MTP used in the experiments satisfy the conditions?  Please justify the numerical implementation of MTP, including the assumption of infinite number of moments, the use of $\sigma(\Lambda^k)$ and how the metric properties are affected.

2. In Corollary 1, if both feature projection and MTP are injective, why the data point projection, as the sum of them, is also injective? Could the authors explain why the data point projection is a composition of Radon transform projection and the MTP?

**Essential References Not Discussed:**

Literature was well reviewed.

**Experimental Designs Or Analyses:**

Experiments are valid and sound.

**Methods And Evaluation Criteria:**

Experiments are supportive and sufficient. However, can you explain more about the feature projection choices for each of the dataset?

**Other Comments Or Suggestions:**

Typos:
1. $\Lambda\subset \mathbb{N}$ instead of $\Lambda\in \mathbb{N}$
2. In Preliminaries, at line 133, $n$ should be subscript.
3. In Algorithm 1, at line 683, $i=1'$ should be $i'=1$

**Other Strengths And Weaknesses:**

The proposed method using MTP to project labels onto 1D slices is novel and interesting. The computation speed gain is significant.

**Questions For Authors:**

1. In Definition 3, do feature projection and MTP have to share the same $\psi^{(1)}$?

**Relation To Broader Scientific Literature:**

A fast dataset distance can be used in tasks that involve comparisons of datasets, such as transfer learning,

**Theoretical Claims:**

The injectivity conditions of MTP, metric properties of s-OTDD and the numerical approximation bounds are checked and correct.

---

> ### Author Rebuttal · Authors · 2025-03-31
>
> We appreciate the time and constructive feedback of the reviewer.  We would like to extend the discussion with the reviewer as follows:
>
> **Q6**. Proposition 1 gives two conditions for the MTP to be injective, and the metric properties of s-OTDD are based on the injectivity of MTP. Does the MTP used in the experiments satisfy the conditions? Please justify the numerical implementation of MTP, including the assumption of infinite number of moments, the use of $\sigma(\Lambda^k)$ and how the metric properties are affected.
>
> **A6**.  Checking existence of all moments of a high-dimensional distribution is impractical due to a high computational cost. Moreover, we only observe random samples from the unknown distribution which makes the estimation of moments are also random. Therefore, in experiments, we implicitly assume that the underlying distributions of datasets have infinite moments. This assumption is motivated by the fact that one-dimensional projection becomes a Gaussian distribution in high-dimension [1,2] and Gaussian distribution has infinite number of moments. We then use zero truncated Poisson distribution (as reported at line 309-310) for the value of moments since it is a distribution over infinite countable natural numbers. If the underlying distributions of datasets do have infinite moments, MTP is injective and s-OTDD is a metric.  We would like to recall that s-OTDD is still a pseudo distance without injectivity of the MTP i.e., s-OTDD satisfies the triangle inequality, symmetry, non-negativity, and one direction of identity (is 0 when two datasets are the same). Therefore, s-OTDD is still a meaningful discrepancy for datasets as demonstrated through our experiments.
>
> [1] Fast approximation of the sliced-Wasserstein distance using concentration of random projections, NeurIPS 2021, Nadjahi et al.
>
> [2] Asymptotics of Graphical Projection Pursuit, The Annals of Statistics, 1984 Diaconis et al.
>
>
> **Q7.** In Definition 3, do feature projection and MTP have to share the same $\psi^{(1)}$?
>
> **A7.** It is actually a typo in our Eq. (9).  The summation in the second term should involve $\psi^{(i+1)}$. We have fixed this typo in the revision. In particular,  we have the following revised definition of the data point projection
>
> $\mathcal{DP}^k_{\psi,\theta,\lambda,\phi}(x,q_y) = \psi^{(1)} \mathcal{FP}_\theta(x) $
>
> $+ \sum_{i=1}^k\psi^{(i+1)} \mathcal{MTP}_{\lambda^{(i)},\phi}(q_y)$,
>
> where $\psi=(\psi^{(1)},\psi^{(2)},\ldots,\psi^{(k+1)}) \in \mathbb{S}^k, \theta \in \Theta, \lambda =(\lambda^{(1)},\ldots,\lambda^{(k)}) \in \Lambda^{k}, \phi \in \Phi$.
>
> **Q8**. In Corollary 1, if both feature projection and MTP are injective, why the data point projection, as the sum of them, is also injective? Could the authors explain why the data point projection is a composition of Radon transform projection and the MTP?
>
> **A8**. Thank you for your questions. From the revised definition in **A7.** let $\mathcal{FP}_\theta(x)=t_1$
>
> and
> $\mathcal{MTP}_{\lambda^{(i)},\phi}(q_y)=t_i$ for $i=2,\ldots,k$, we can rewrite the data point projection as:
>
> $\mathcal{DP}^k_{\psi,\theta,\lambda,\phi}(x,q_y) = \psi^{(1)} t_1+ \sum_{i=2}^k\psi^{(k)} t_i = \psi^\top t = \mathcal{R}_\psi(t)$,
>
> where $\psi=(\psi^{(1)},\ldots,\psi^{(k)})$, $t=(t_1,\ldots,t_k)$, and $\mathcal{R}_\psi(t)$ is the Radon Transform of $t$ with projection parameter $\psi$ as defined at line 162. Therefore, the data point projection is nothing but the Radon Transform of the stacked output of the feature projections and moment transform projections. Since the composition of injective functions is also injective, the data point projection is injective given the injectivity of the feature projection and the moment transform projections.
>
> **Q9**. Can you explain more about the feature projection choices for each of the dataset?
>
> **A9.** As written at line 194, we choose the feature projection based on the prior knowledge about the feature space. For example, if we believe that the feature space is an Euclidean space,  we can use the Radon projection (linear projection). If we work with a manifold, we can use geodesic projection [3]. If we work with images, we can use the convolution projection [4]. In our experiments, for the NIST dataset and the text classification dataset,  we apply linear projection by default. For image data, we use convolutional projection. We conducted a new comparison between linear and convolutional projections, available at [https://imgur.com/a/1NVv5AC](https://imgur.com/a/1NVv5AC). The results show that convolution-based projections not only require fewer projections but also tend to exhibit a stronger positive correlation.
>
> [3] Sliced-Wasserstein Distances and Flows on Cartan-Hadamard Manifolds, JMLR, 2025, Bonet el al.
>
> [4] Revisiting Sliced Wasserstein on Images: From Vectorization to Convolution, NeurIPS, 2022, Nguyen et al.
>
> **Q10.** Typos
>
> **A10.** We have fixed them in the revision.

---

> > ### Comment · Reviewer_FRUy · 2025-04-05
> >
> > I appreciate the authors' response. All my concerns were addressed. I will adjust the score. It would also be great if the authors could add the clarification about the injectivity of MTP in practice and the s-OTDD being a pseudo-metric in the paper.

---

> > > ### Author Response · Authors · 2025-04-05
> > >
> > > We would like to thank the reviewer for increasing the score to 4. We will include all the discussion to the revision of the paper. Please let us know if you still have other questions.
> > >
> > > Best regards,
> > >
> > > Authors,

---

### Official Review · Reviewer_VKhK · 2025-03-24

**Overall Recommendation:** 3

**Summary:**

This paper is a straightforward application of sliced OT on OT-based dataset distances. The main novelty is Moment Transform Projection by which the authors could project dataset labels to scalars, enabling the use of Radon transformation and hence sliced Wasserstein distances. Authors then follow standard procedures to prove the metric properties and approximation error of the sliced OT dataset distance. Empirical results from various experiments show that the sliced OT dataset distance is both efficient and effective.

**Claims And Evidence:**

Yes.

**Essential References Not Discussed:**

References are sufficient. I'd add work by Ho et al who contributed to some theoretical findings around sliced OT or Kolouri et al. as an example of the applications of sliced OT but that's not necessary.

**Experimental Designs Or Analyses:**

Results from empirical experiments are convincing in verifying the claim that the proposed sliced OTDD is much more efficient than traditional OTDD, and the approximation error is acceptable.

**Methods And Evaluation Criteria:**

Yes.

**Other Comments Or Suggestions:**

None.

**Other Strengths And Weaknesses:**

A weakness of the proposed sliced OTDD is the extra hyper-parameter — the number of moment $\lambda$ as in (7). In the code, it’s set to 5. I didn’t find it discussed in the paper except in 246 where authors only briefly mentioned the principle of choosing $\lambda$. It’s impact to the projection and hence the distances is unknown as well.

**Questions For Authors:**

The $\lambda$-th scaled moment as defined in (7) is not how I usually "scale" a moment. What's the rationale behind a constant scaler?

Please discuss the impact of $\lambda$ and provide more details for A.1.

**Relation To Broader Scientific Literature:**

The paper is a sweet overlap between sliced OT and OT-based dataset distance. It's a natural application of sliced OT to another one of OT's applications. The main contribution of the paper is not to identify the overlap or the technical difficulty, but to lay the necessary theoretical foundation in a timely manner so that on one hand researchers could advance their work on top of this work and on the other hand people can use OT dataset distance in practice.

Sliced OT until now is a somewhat well-studied subject after Bonneel et al., 2015, Nietert et al., 2022, and others. And as the authors listed, Alvarez-Melis, D. and Fusi, N. 2020 is a milestone for OT-based dataset distance. It totally makes sense to fill gap between the two subjects.

**Theoretical Claims:**

I checked all the proofs. Proofs in A.1. jump too much between lines. Please expend them and add intermediate steps.

---

> ### Author Rebuttal · Authors · 2025-03-31
>
> First, we would like to thank the reviewer for the review and feedback. We would like to answer questions from the reviewer as follows:
>
> **Q3.** A weakness of the proposed sliced OTDD is the extra hyper-parameter - the number of moment $\lambda$ as in (7). In the code, it’s set to 5. I didn’t find it discussed in the paper except in [246] where authors only briefly mentioned the principle of choosing $\lambda$. Its impact on the projection and hence the distances is unknown as well.
>
> **A3.** Thank you for your insightful comments.  We would like to recall that the number of moments is denoted as $k$ and we did report the choice of $k=5$ at line 307-308 in the paper. For the parameter $\lambda$, it is the order of a moment. We would like to extend the discussion on both $k$ and $\lambda$ as follows:
>
> For $k$, it plays the role of selecting the number of moments for the label projection. In particular, the higher value of $k$, the more moments information of the label distribution is gathered into the data point projection. For $\lambda$, it plays the role of choosing information of distributions to extract. For example, $\lambda=1$ leads to the information of centerness while $\lambda=2$ leads to the information of spread. In the paper, $\lambda_i$ for $i=1,\ldots,k$ are random which followed the zero-truncated Poisson distribution with the rate parameter equals $i$ i.e.,  the mean  $\mathbb{E}[\lambda_i]= (ie^i)/(e^i-1) \approx i$. It means that we try to capture (in high probability) the first $k$ moments of the label distribution. In the case, where we know the true number of moments of the label distribution $\lambda^\star$, we can set $k=\lambda^*$ to capture all information of the label distribution. Nevertheless, checking existence of moments is expensive since we only access to samples from the dataset generating distribution and  the size of datasets is also large. Therefore, $k$ becomes a hyperparameter. However, choosing $k$ is not a hard problem since we know that we want to have as big $k$ as possible. In practice, we can start with a big $k$ and check if the value of s-OTDD exists (not overflow). If s-OTDD does not exist, you can reduce the value of $k$ via a binary search rule to have a choice of $k$. To avoid such searching algorithm, we recommend to choose $k=5$ due to the concentration of random projection. In particular, we know that one-dimensional projection becomes a Gaussian distribution in high-dimension [1,2]. Therefore, two moments are enough to capture the information of a Gaussian distribution. Since we have finite dimension, using extra 3 moments (5 in total) must be enough.
>
> To verify the above hypothesis, we have conducted additional ablation studies in *NIST Adaptation where we vary $k\in \\{1,2,3,4,6\\}$, resulting visually at [https://imgur.com/a/NCcqYgo](https://imgur.com/a/NCcqYgo). We see that increasing $k$ leads to better correlation to  the performance gap. Nevertheless, we found that when passing $k=3$, the correlation does not increase fast as when $k<3$. Also, we found that when setting $k$ to be too large, s-OTDD is not computable due to numerical issue i.e., overflow which might be due to the non-existence of the higher moments. We will add this discussion and the additional experiments to the revision.
>
> [1] Fast approximation of the sliced-Wasserstein distance using concentration of random projections, NeurIPS 2021, Nadjahi et al.
>
> [2] Asymptotics of Graphical Projection Pursuit, The Annals of Statistics, 1984 Diaconis et al.
>
> **Q4.**. The $\lambda$-th scaled moment as defined in (7) is not how I usually "scale" a moment. What’s the rationale behind a constant scaler? Please discuss the impact of $\lambda$.
>
> **A4.** The reason we scale $\lambda$-th moment by $\lambda!$ is to make sure the output of the data point projection is not biased toward high order moments. In particular, in the definition of the data point projection (Definition 3), the data point projection is a weighted average of the feature projection and multiple moment transform projections with different values of $\lambda$. If we do not scale the moment, the value of the data point projection will be entirely based on the value of the moment transform projection with highest value of $\lambda$. However, we know that each moment captures different information of the distribution e.g., the first moment captures the information about center, the second moment captures the information about the spread. Therefore, scaling the moment value makes the contribution of all moments "approximately" equal. The reason we say that they are "approximately" equal is that we actually imply the priority for small order moments since the factorial normalizing constant goes faster in limit than the exponential function.  We will include this discussion in the revision of the paper.
>
> **Q5.** provide more details for A.1.
>
> **A5.** Thank you for your comments.  We will expand the proofs by adding intermediate steps.

---

### Official Review · Reviewer_1ZM5 · 2025-03-24

**Overall Recommendation:** 4

**Summary:**

This paper tackles the Dataset Distance problem with a proposed sliced optimal transport dataset distance (s-OTDD) method.
The core module is called Moment Transform Projection (MTP), mapping a label (represented as a distribution over features) to a real number. Then, s-OTDD is defined as the exptected Wasserstein distance between the projected distributions, and calculated by leveraging the closed-form one-dimentional optimal transport, i.e. sliced Wasserstein distance.
With random projection, s-OTDD shows (near-)linear property for the Dataset Distance problem.

Experiments are conducted on various benchmarks, and the main comparison method is OTDD(exact). OTDD can be regarded as the gold standard or upper bound of the s-OTDD algorithm. The results shows good correlation between OTDD and s-OTDD, while the compuation time is magnitude of faster.

Moreover, several theoretical results are given: injective property, s-OTDD is a valid metric, and the approximation error.

## update after rebuttal
I appreciate the authors' response. All my questions and concerns are addressed by the rebuttal.
I am happy to keep my score as 4: Accept.

**Claims And Evidence:**

The s-OTDD is a fast and effective approximation of the OTDD dataset distance method. This is verified by the theorecial proof (valid metric and approximatoon error), as well as experimental comparison on various benchmark dataset (correlation w.r.t. OTDD (exact)).

s-OTDD is a valid distance, which is proved in Proposition 2.

**Essential References Not Discussed:**

To my knowledge, I do not know other relevant references.

**Ethical Review Concerns:**

No ethical concern,

**Experimental Designs Or Analyses:**

Overall, the experiments are sound and effective.
(1) Various benchmarks and tasks are conducted, e.g. image classification and text classification on MNIST, CIFAR, TinyImageNet, AG News, DBPedia, Yelp Reviews, Amazon Reviews, and Yahoo Answers.
(2) The important baseline and standard distance OTDD(exact), as well as a bunch of other approximation distances are compared.
(3) Running time comparison is given w.r.t dataset size.
(4) Parameter analysis are given w.r.t number of projections.

**Methods And Evaluation Criteria:**

Yes.
The proposed method mainly leverages the one-dimensional closed-form property of the optimal transport, which also translates to the sliced-Wasserstein distance.
Based on this, the proposed method designed Moment Transform Projection to ensure the injective feature projection. Then it applies data point projection.
Finally, Sliced Optimal Transport Dataset Distance is approximated random projection over four parameters ($\psi, \theta, \lambda, \phi$) sampled from corresponding spaces.

Evaluation metrics mainly include (1) distance correlation w.r.t. OTDD and other dataset distances (2) processing time w.r.t. dataset size. Overall, these metrics can capture the effectiveness and efficiency of s-OTDD.

**Other Comments Or Suggestions:**

Typos.
The line below Eq. (8), empirical distribution misses the variable.

**Other Strengths And Weaknesses:**

The main Strengths and Weaknesses have already listed in the above sections.

**Questions For Authors:**

In Eq. (9), are the first $\psi^{(1)}$ in the first term, the same as the $\psi^{(k)}$ with $k=1$ in the second term?

**Relation To Broader Scientific Literature:**

This paper mainly provides an effective and much faster dataset distance, which improves the OTDD (Alvarez-Melis & Fusi, 2020).

Through theoretical proof and experimental comparisons, the proposed distance s-OTDD approximates OTDD and also is much faster in speed w.r.t dataset size.

**Theoretical Claims:**

Proposition 1. Existence of projected scaled moments. The proof is given in Appendix A.1

Proposition 2. s-OTDD is a valid metrics. The proof is given in Appendix A.2

Proposition 3. The approximation error. The proof is given in Appendix A.3

I went through all the proof and did not find theoretical issue. Because this is an emergent review, I might miss some detail due to limited review time.

---

> ### Author Rebuttal · Authors · 2025-03-31
>
> First, we would like to thank the reviewer for the time and the feedback. We extend the discussion as follows:
>
> **Q1.** The line below Eq. (8), empirical distribution is missing the variable.
>
> **A1.** Thank you for pointing out the typo. We have fixed the typo in the revision.
>
> **Q2.** In Eq. (9), is the first $\psi^{(1)}$ in the first term, the same as the $\psi^{(k)}$ with $k = 1$ in the second term?
>
> **A2.** It is actually a typo in our Eq. (9). The summation in the second term should involve $\psi^{(i+1)}$. We have fixed the typo in the revision. In particular,  we have the following revised definition of the data point projection:
>
> $\mathcal{DP}^k_{\psi,\theta,\lambda,\phi}(x,q_y) = \psi^{(1)} \mathcal{FP}_\theta(x) $
>
> $+\sum_{i=1}^k \psi^{(i+1)} \mathcal{MTP}_{\lambda^{(i)},\phi}(q_y)$
>
> where $\psi=(\psi^{(1)},\psi^{(2)},\ldots,\psi^{(k+1)}) \in \mathbb{S}^k, \theta \in \Theta, \lambda =(\lambda^{(1)},\ldots,\lambda^{(k)}) \in \Lambda^{k}, \phi \in \Phi$.
>
> We would like to elaborate more on Corollary 1 with the revised definition. Let $\mathcal{FP}_\theta(x)=t_1$
>
> and $\mathcal{MTP}_{\lambda^{(i)},\phi}(q_y)=t_i$ for $i=2,\ldots,k$, we can rewrite the data point projection as:
>
> $\mathcal{DP}^k_{\psi,\theta,\lambda,\phi}(x,q_y) = \psi^{(1)} t_1+ \sum_{i=2}^k\psi^{(k)} t_i = \psi^\top t = \mathcal{R}_\psi(t)$,
>
> where $\psi=(\psi^{(1)},\ldots,\psi^{(k)})$, $t=(t_1,\ldots,t_k)$, and $\mathcal{R}_\psi(t)$ is the Radon Transform of $t$ with projection parameter $\psi$ as defined at line 162. Therefore, the data point projection is simply the Radon Transform of the stacked output of the feature projections and moment transform projections. Since the composition of injective functions is also injective, the data point projection is injective given the injectivity of the feature projection and the moment transform projections.

---

### Decision · Program_Chairs · 2025-05-01

**Decision:**

Accept (poster)

**Comment:**

The paper introduces s-OTDD, a computationally efficient and model-agnostic dataset distance based on sliced optimal transport, with a novel projection scheme (Moment Transform Projection) enabling the use of 1D Wasserstein distances. The approach is theoretically sound, with clear metric properties and approximation guarantees, and is shown to scale well to larger datasets, including settings where OTDD becomes infeasible. The reviewers appreciated the method’s practicality, its clean and general formulation, and its relevance to tasks such as transfer learning and data augmentation. However, some concerns were raised about the experimental validation. In particular, the evaluation setup largely follows that of prior work on OTDD, with limited exploration of new applications or more complex empirical scenarios. While the rebuttal included an additional experiment on Tiny ImageNet, showing that s-OTDD can operate where OTDD cannot, it remained limited to relatively standard classification settings and subsampled datasets, and did not thoroughly address cases involving, for instance, significant label imbalance, multi-label tasks, or structured prediction. Additionally, while the authors clarified many technical questions, including the injectivity assumptions and hyperparameter choices (such as the number of moments), a more systematic exploration of these design decisions, as well as an expanded discussion of practical limitations (e.g., s-OTDD being a pseudo-metric when injectivity is not guaranteed), would strengthen the final version. Reviewers also noted that, although Pearson correlation with OTDD is a reasonable baseline metric, incorporating rank correlations such as Spearman’s would provide a more robust view of how well s-OTDD aligns with practical downstream performance, especially for transfer tasks, and such metrics were explored only late in the rebuttal. Despite these limitations, the paper offers a timely, well-executed contribution with both theoretical and practical value, and provides a valuable tool for scaling dataset comparison methods to modern data sizes. On balance, while some aspects of the work would benefit from further development and broader validation, I believe the strengths of the work outweigh the weaknesses, and I recommend acceptance.